# The regenerative period of somatosensory nerves is closed by a DCC signaling axis

Jacob Hammer[1,2], Cody J. Smith[1,2]*

1 Department of Biological Sciences, University of Notre Dame, Notre Dame, Indiana, United States of America, 2 The Center for Stem Cells and Regenerative Medicine, University of Notre Dame, Notre Dame, Indiana, United States of America

* csmith67@nd.edu

## Abstract

Tissues and organs have periods of plasticity that close with age. While period closures can lock in tissue architecture and prevent aberrant cellular interactions, they also limit regenerative capacity. These regenerative periods – a timeframe with regeneration capacity – are defined, but the underlying genetic mechanisms that close specific regenerative periods remains critical knowledge that needs expanding. Here, we established zebrafish larvae as a model to study the genetic basis of regenerative period closure. We demonstrated that laser axotomy of the centrally-projecting axons of dorsal root ganglia (DRG) neurons exhibit a robust regenerative period that is closed by 3 days post fertilization (dpf). The closure of the regenerative period corresponds with the rearrangement of glia that express *netrin*, introducing the idea that changes in the DCC-mediated signaling axis could be a genetic and molecular basis closing the regenerative period. To test this hypothesis, we manipulated *dcc*, cAMP, and Rac1 in transgenic animals that label axons and the actin cytoskeleton. Combined with genetic epistasis analysis, we show that altering DCC signaling can re-open the regenerative period, allowing severed axons to regrow into the spinal cord. We show that this increased capacity to reinvade the spinal cord is mediated by growth cone invadopodia. Using calcium reporters and behavioral analysis, we demonstrate that re-opening the regenerative period by manipulating the DCC signaling axis restores the sensory circuit and sensory-specific behaviors. By introducing this genetic basis for regenerative period closure, these results reveal an active suppression process that keeps regenerative periods closed and establishes a new model for future dissection of such periods.

## Author summary

The regenerative capacity of organs and tissues declines with age. The closure of regenerative periods often occurs during early developmental stages

**Data availability statement:** All data related to this manuscript is available in the enclosed material.

**Funding:** Funding for this research was provided by the University of Notre Dame, the Elizabeth and Michael Gallagher Family, the Fitzgerald Family, the Centers for Zebrafish Research and Stem Cells and Regenerative Medicine at the University of Notre Dame, the Indiana Spinal Cord and Brain Injury Research Trust through the Indiana State Board of Health (CJS), the SMART foundation (CJS), and the National Institutes of Health (grant number DP2NS117177-CJS). Awards from Indiana State Board of Health (CJS), the SMART foundation (CJS) and the NIH (CJS) funded salaries for JH and CJS. The funders had no role in study design, data collection and analysis, decision to publish, or preparation of the manuscript.

**Competing interests:** The authors have declared that no competing interests exist.

and severely limits the recovery from injuries and disease. While regenerative periods are defined for many tissues, the underlying molecular mechanisms that close most regenerative periods remain limited. Here, we address this by investigating the regenerative period of a subset of neurons that relay sensory information from the skin into the spinal cord. By establishing zebrafish as a model to study regenerative periods, we demonstrate that the regenerative period of sensory nerves rapidly declines within 24 hours during development. Our work highlights that the regenerative period for these nerves is closed due to an innate re-organization of supportive cells in the tissue, which likely overactivates a molecular signaling pathway. While this signaling pathway functions during the initial construction of the nerve, its continued activation limits the regeneration of the nerve. We show that by manipulating this pathway we can re-open the regenerative period of these essential nerves. Collectively, our work demonstrates an active suppression mechanism that closes the regenerative period.

## Introduction

Cellular plasticity is a hallmark of developing organs that declines with age [1]. Periods of cellular plasticity are closed in context-dependent ways, varying by organ, cell-type, and physiological condition [1–3]. Such closure of plasticity periods helps to secure tissue architecture during development. While this ensures organs can shift from developmental construction to a functioning organ, closing the plasticity period can have negative consequences like stymieing regeneration in injured or diseased states [4–7]. Understanding the conserved mechanisms that govern the opening and closing of these developmental plasticity periods is critical to understand the fundamental processes of tissue morphogenesis, functional maturation, and the etiology of developmental disorders.

The nervous system provides a powerful lens through which to examine the principles governing developmental plasticity [7,8]. Within this context, critical periods – defined as discrete windows of time during which neural circuits exhibit heightened sensitivity to experience-dependent refinement – have been extensively studied [9–13]. These critical periods are orchestrated by specific genetic mechanisms operating in a context-dependent manner. For instance, critical periods of distinct brain regions in mammals occur at different developmental stages, highlighting the spatio-temporal specificity of this phenomenon [9,10,14]. Notably, the existence and general features of critical periods are conserved across phylogeny, suggesting fundamental developmental principles at play. Their initiation, maintenance, and eventual closure are governed by intrinsic neuronal programs and intricate interactions with glial cells [15–18]. Aberrant regulation of critical period closure has been implicated in various neurodevelopmental disorders, underscoring the importance of their precise timing for proper brain function [19,20].

The concept of critical periods in neural development parallels regenerative periods. These define specific developmental windows during which organs and tissues

possess a robust capacity for repair and regeneration following injury. Similar to the precisely timed closure of critical periods, the cessation of regenerative potential represents a significant developmental transition. For example, the mammalian heart can robustly regenerate in neonates at postnatal day 1 (P1) to P4, but its regenerative capacity quickly declines and is lost by P7 [21]. The nervous system similarly exhibits closure of regenerative periods [22]. The spinal cord can partially regenerate in mammals from P1-P4 but this capacity declines with age to eventually exhibit low levels of regeneration in adulthood [23]. In zebrafish, somatosensory neuron regeneration of axons that innervate the skin diminishes with age [24]. Similarly, afferent spinal somatosensory nerves of the dorsal root ganglia (DRG) can regenerate in neonatal mammals but fail to regenerate shortly after birth, indicating that these centrally-projecting nerves are also governed by a tightly regulated regenerative period [25–27]. Such an injury during the birth process, such as obstetrical brachial plexus injuries, can lead to permanent sensorimotor defects in children [28,29]. The inability of organs to regenerate can be explained by precise genetic and cellular mechanisms that are natural to the native tissue or non-permissive injury responses [25,30–33]. The classic example of this is the glial scar which only occurs as a result of spinal cord injury and is thought to limit regeneration of surviving axons [34–37]. Nonetheless, the decline in regenerative capacity is relevant in a clinical setting, limiting the recovery potential for infants, children, and adults from injury or disease states.

The centrally-projecting axons of DRG, which relay vital somatosensory information into the central nervous system (CNS), exhibit a remarkable decline in regenerative capacity with age in mammals. The reduced regenerative capacity is driven by both changes in the neurons and the surrounding glial landscape. For example, it has long been understood that regeneration of mammalian centrally-projecting axons is inhibited by glial-derived signals and physical scars, as well as neuronal intrinsic mechanisms like the downregulation of specific pro-regenerative factors [38–41]. After injury, centrally-projecting axons of the DRG must navigate in the periphery and then invade into the spinal cord boundary to regenerate [42]. While these distinct phases and the mechanisms regulating them have not been extensively dissected in regeneration, timelapse imaging in zebrafish demonstrate that developing pioneer DRG axons use filopodia to initially navigate in the periphery and then shift to invadopodia to cross the spinal cord boundary [43,44]. Invadopodia are specialized actin-based structures that penetrate structural boundaries via physical and chemical processes [42,43,45–48]. Unlike in development, studies in zebrafish demonstrate that regenerating central axons do not form stable invadopodia at the invasion site and thus fail to re-enter and reconnect with spinal circuits [42]. While these studies point to inhibitory factors that reduce regeneration capacity, the genetic mechanisms that close the regenerative period remain mostly unknown.

Here, we developed a zebrafish model to reveal the genetic mechanisms that close regenerative periods in the developing animal. Using laser-induced axotomy at different developmental stages, we reveal that the regenerative period for centrally-projecting axons is open at 2 days post fertilization (dpf) but rapidly closes within 24 hours. This closure of the regenerative period corresponds with rearrangement of Netrin-expressing glia around the DRG axon. With genetic and molecular manipulations of the Netrin receptor, DCC (Deleted in Colorectal Cancer), and its signaling pathway, which includes cAMP and Rac1, we can re-open the regenerative period and thereby promote regeneration of these afferent nerves. Our results indicate that the regenerative period is closed by the active suppression of invadopodia by the DCC signaling pathway. Behavioral analysis, combined with analysis of calcium dynamics, supports the idea that re-opening the regenerative period can ensure functional recovery after afferent spinal somatosensory axotomy. Together our results reveal the genetic mechanisms of regenerative period closure and introduce a new genetic model to explain the marked decline in regeneration that occurs with age.

## Results

### Centrally-projecting DRG axons in zebrafish exhibit regenerative period closure

To develop a model to study the genetic mechanisms of regenerative periods, we first determined if developing zebrafish exhibit reduced regenerative capacity with age. We investigated the regenerative capacity of centrally-projecting DRG axons because of their established regenerative period in mammals and clinical relevance to brachial plexus injuries in

humans. To visualize the DRG and its central axon bundle we expressed GFP using *ngn1* promoter elements and visualized the glial limitans (spinal cord boundary) with mCherry driven by the *gfap* promoter in *Tg(ngn1:GFP);Tg(gfap:NTR-mCherry)* animals (Fig 1A and 1B) [49,50]. We assayed re-invasion and re-entry with a published analysis protocol, orthogonal displacement quantifications, where the dorsal tip (growth cone) of the GFP+ regenerating axon, if present, was located in the orthogonal image relative to the mCherry+ glial limitans (Fig 1A) [43]. Plotting the fluorescent intensity profile at the tip of the GFP+ axon allowed us to determine if the regenerated axon was medial or lateral to the mCherry+ glial limitans (Fig 1C). Utilizing our confocal laser-pulse lesioning system, we axotomized the central axon bundle of a single DRG per animal in 10–12 animals at 2, 3 and 5 dpf. We then re-imaged these DRG 24 hours post-injury (hpi) (Fig 1B) to perform the orthogonal displacement quantifications (Fig 1C) and establish when the regenerative period closes.

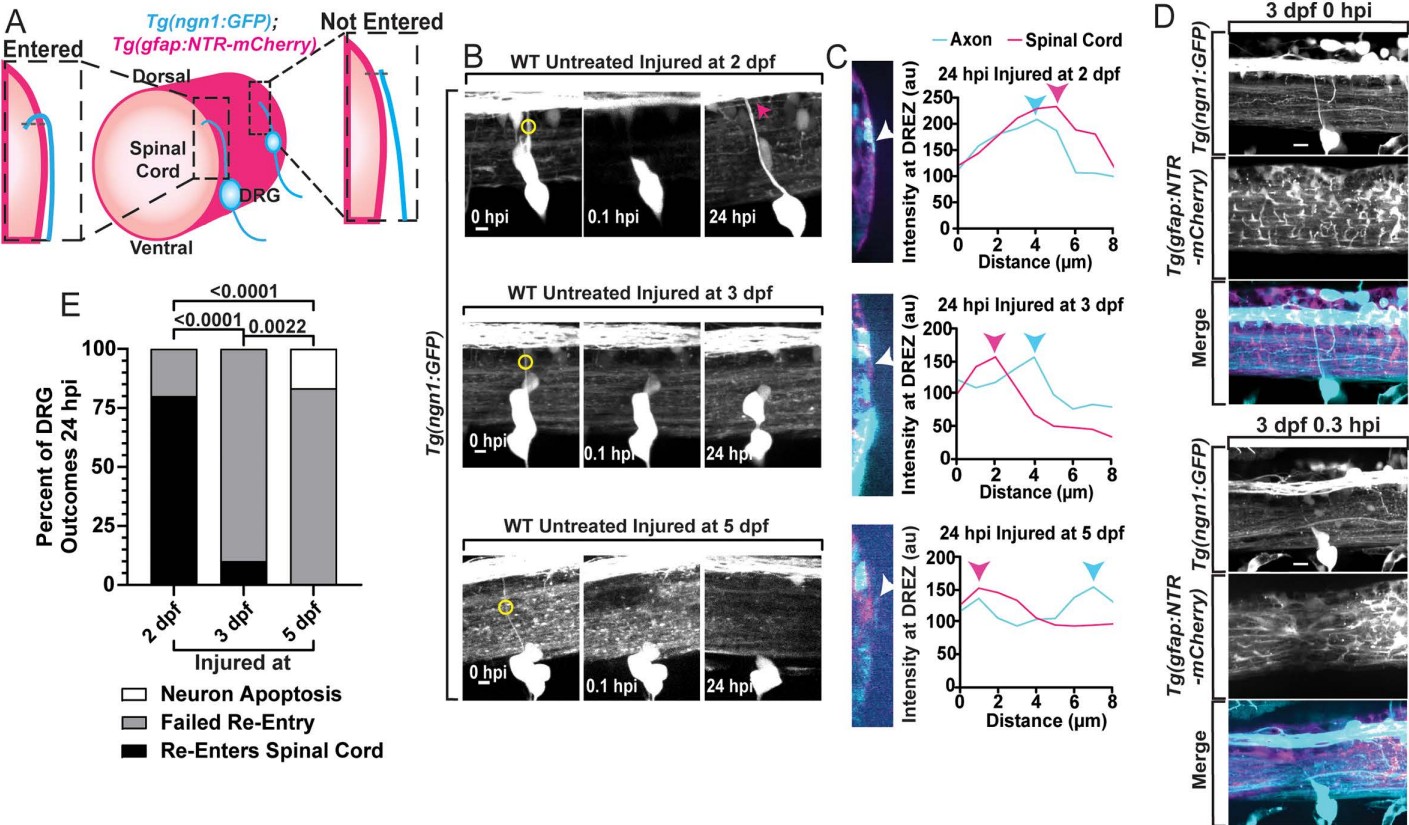

**Fig 1. Centrally-projecting DRG axons experience a decline in regenerative capacity between 2 and 3 dpf. A)** A schematic of orthogonal displacement quantifications. A line is drawn across where the distal tip of the axon and spinal cord boundary meet to quantify the fluorescent profiles of each marker. This quantification is used to determine if previously axotomized DRG central axons regenerate by scoring re-entry into the spinal cord. **B)** Max z-projection images and of 2, 3, and 5 dpf *Tg(ngn1:GFP)* DRG axons pre-injury (0 hpi), post-injury (0.1 hpi) and 24 hpi (scale bar = 5 μm). Yellow circles on the pre-injury image indicate where the laser lesioning system was targeted. **C)** Orthogonal sideview images from the 24 hpi captures from each age group (injured at 2, 3 and 5 dpf). These images depict the right side of the spinal cord (pink) and the tip of the regenerative axon (cyan). The adjacent graphs are the orthogonal displacement quantifications, plotting the fluorescent profiles of the axon and spinal cord boundary, displaying if the axon has re-entered the spinal cord boundary (axon fluorescent peak is left or inside of the spinal cord boundary fluorescent peak) or failed to re-enter (axon peak is right/outside of spinal cord boundary peak). White arrowheads point to the distal tip of the axon where these fluorescent profiles were obtained. Pink and cyan arrowheads point to the fluorescent peak of the spinal cord boundary and axon, respectively. **D)** Representative max projection images displaying the DRG and spinal cord (glial limitans) before and after axotomy. Post-injury images display that axotomy limits excessive damage to surrounding tissues (scale bar = 10 μm). **E)** 24 hpi outcomes of the DRG central axons when injured at 2 dpf (n = 10 animals, 1 central axon per animal), 3 dpf (n = 10 animals), and 5 dpf (n = 12 animals). Comparison of the percentages of re-entry and failed re-entry were made between age groups using Fisher's Exact tests. Raw data information for this figure can be found in the S1 Data.

This lesioning system targets select diffraction-limited z-planes with scalable laser pulses to minimize injury to surrounding tissues. We confirmed that such injuries cause a complete transection of the centrally-projecting nerve while limiting damage to the surrounding area (Fig 1D). Zebrafish DRG pioneer axons extend, invade and bifurcate in the spinal cord between 40–48 hpf, while also adding additional axons in the nerve after the initial pioneer axon invades. Injured DRG axons must re-navigate to the dorsal root entry zone (DREZ) and then re-invade across the spinal cord boundary to re-enter the CNS and re-connect with spinal circuits. From the orthogonal displacement quantification analyses, we observed that 80% (n = 10) of axons injured at 2 dpf were medial to the glial limitans by 24 hpi, indicating successful re-entry/regeneration into the spinal cord. In contrast, only 10% (n = 10) of axons injured at 3 dpf had regenerated across the glia limitans (Fig 1E). We could not detect regenerated axons medial to the glial limitans after injuries performed at 5 dpf (n = 12). Consistent with previous reports, once an axon failed to re-invade it retracted to the cell soma [42]. These data indicate that zebrafish centrally-projecting DRG axons experience a regenerative period that rapidly closes in the 24 hours between 2 and 3 dpf and thereby establishes zebrafish as a model to investigate the mechanisms that close the regenerative period.

## Glial cells re-organize during regenerative period closure

To determine the cellular mechanisms that close the regenerative period, we first explored if there were structural changes in the tissue between 2 and 3 dpf. Several glial cell populations, including Schwann cells, satellite glial cells, oligodendrocyte progenitors and astrocytes have been shown to respond to injury [51–55]. To determine the differential response of glia between 2 and 3 dpf, we used *Tg(ngn1:GFP);Tg(sox10:nls-Eos)* animals to visualize the GFP⁺ DRG neurons and centrally-projecting axons and photoconverted *sox10*⁺ glial nuclei surrounding the DRG (Fig 2A) [56]. In timelapse imaging after axotomy, we calculated the migration distances of the axon (based on growth cone position) and the dorsal-most glial nucleus surrounding the DRG soma (Fig 2B and S1–S2 Movies). These migration distance measurements started when regenerative axons began their re-extension at 11.6 hours (5 min intervals, 140 timepoints, Fig 2B). The glial nuclei in 3 dpf animals were consistently positioned more dorsal to the DRG compared to 2 dpf animals (Fig 2A–2B). Furthermore, glia migrated dorsally throughout the timelapse in 3 dpf animals, with an average position of 10.85 ± 2.801 μm (SEM, n = 7) dorsal from the dorsal border of the DRG neuron over the 140 timepoints following axon re-extension (Fig 2B–2C and S2 Movie). In stark contrast, the average glial nuclear position in 2 dpf animals was 5.27 ± 3.049 μm dorsal to the DRG (SEM, n = 7) (Fig 2C). This indicates that glia surrounding the DRG differentially re-organize and undergo a significant dorsal migration in 3 dpf animals.

This glial re-organization could be a result of axotomy or a part of an innate developmental process. To distinguish between these possibilities, we performed mock injuries (outside the glial nuclei and central axon, Fig 2A) in 3 dpf animals and observed that glia were an average of 10.94 ± 3.150 μm dorsal to the DRG (SEM, n = 6, Fig 2C) over the 140 measured timepoints. This migration was comparable to axotomized 3 dpf animals, consistent with the idea that axotomy itself is not required for glial re-organization. Quantifications of the total dorsal migration distances (t0-t140) in 3 dpf animals that were either mock-injured (7.209 ± 1.617 μm (n = 6)) or injured (8.081 ± 0.8504 μm (n = 7)) showed that, when compared with 2 dpf injured animals, a similar glial re-organization does not occur at 2 dpf (2.119 ± 0.9025 μm (n = 7))(Fig 2D). The simplest explanation for this data is that glia surrounding the DRG inherently re-organize at 3 dpf along the central DRG axon, which intriguingly coincides with regenerative period closure.

To address if glia surrounding the DRG and central axon impact the regenerative period, we used the same laser pulse lesioning settings to ablate neighboring/dorsal glial nuclei in 3 dpf animals prior to central axon axotomy (1 DRG per animal) (Fig 2E). We observed that 33.33% (n = 27 animals) of DRG central axons regenerated by 24 hpi in glia-ablated animals, while only 11.11% (n = 18 animals) regenerated in animals that did not undergo glial ablations (p = 0.0003, Fisher's Exact test) (Fig 2F). Furthermore, to rule out the possibility that non-specific photoablation, rather than specific glial ablation, induced regeneration, we performed mock glial ablations targeted outside the glial nuclei prior to central axon

 

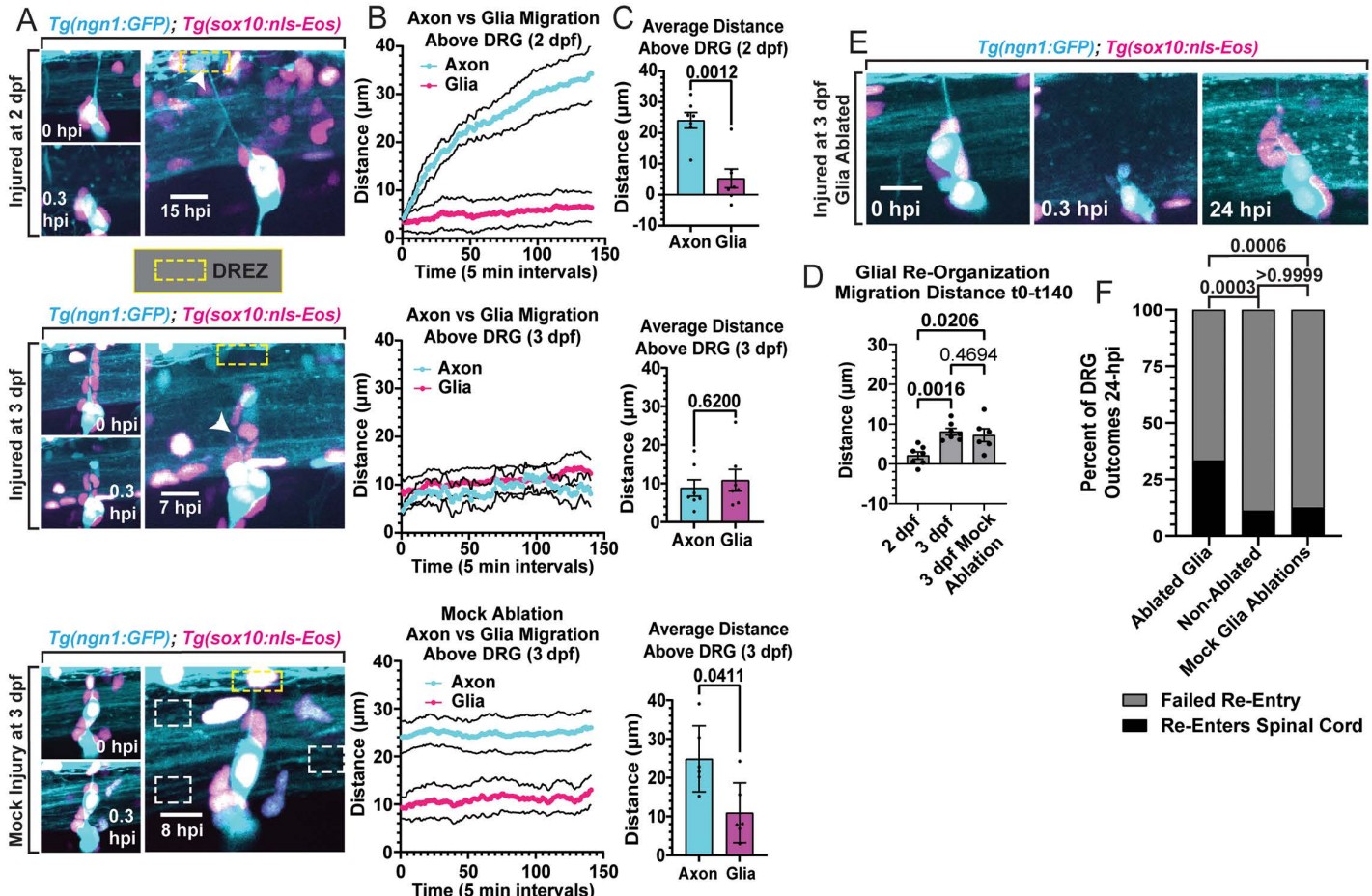

**Fig 2. Glial cells surrounding the DRG re-organize during regenerative period closure. A)** Max projection images of DRG and centrally-projecting axons (cyan), and glial nuclei (pink) of *Tg(ngn1:GFP);Tg(sox10:nls-Eos)* animals pre-injury/pre-mock injuries (0 hpi), following injury/mock injury (0.3 hpi), and during the 24 hpi timelapse of 2 dpf, 3 dpf, and mock ablated 3 dpf animals. Yellow dashed boxes indicate the location of the DREZ. White arrowheads indicate the regenerating central axons' location, and white dashed boxes indicate the location of the mock injury (scale bars = 10 μm). **B)** Line graphs displaying the migration distances (± SEM) of the axon (cyan) and the dorsal-most glial nuclei (pink) surrounding the DRG for 140 timepoints (5 min intervals) after axon re-extension begins or at 4 hour post mock ablation (n = 7 axons and glial nuclei per group in injured animals, n = 6 for mock group). **C)** Quantification of the average distance of the axon and dorsal-most glial nucleus (± SD) over the 140 analyzed timepoints, measured from the top of the DRG. Comparisons between average axon and glial migration distances were made with Mann-Whitney T-Tests (n = 7 per injured group and n = 6 for the mock group). **D)** Total migration distances of glial nuclei per group (± SEM), calculated by subtracting the x, y positions of the dorsal-most glial nuclei at t140 from t0 over the 140 analyzed timepoints per animal (n = 7 per injured group and n = 6 for mock group). The migration distances were compared with Kruskal-Wallis One-Way ANOVA (p = 0.05). **E)** Representative images of 3 dpf *Tg(ngn1:GFP);Tg(sox10:nls-Eos)* animals where the neighboring and dorsal glial nuclei surrounding the DRG were ablated prior to central axon axotomy. Panels show pre-injury and ablation (0 hpi), post ablation and axotomy (0.3 hpi), and 24 hpi with a regenerated central axon (scale bar = 10 μm). **F)** Post-injury outcomes in 3 dpf *Tg(ngn1:GFP);Tg(sox10:nls-Eos)* animals where glial cells were ablated, not ablated, or mock ablated (lesioning targeted outside the nuclei). Fisher's Exact tests compared percentages of re-entry and failed re-entry outcomes between groups. Raw data information for this figure can be found in S1 Data.

axotomy. These mock ablations had no effect on regenerative capacity (12.5% re-entered n = 8 animals, Fig 2F), similar to non-glial ablated animals (p > 0.9999, Fisher's Exact test). These collective results support a hypothesis in which the re-organization of glia surrounding the DRG in 3 dpf animals takes part in closing the regenerative period for centrally-projecting DRG axons.

### *netrin1b* increases in glial cells dorsal to DRG during regenerative period closure

To identify molecular mechanisms that regulate regenerative period closure, we searched for transcripts expressed in glia that surround the DRG neurons. It was previously demonstrated that the invasion of developmental pioneer DRG central axons across the DREZ is regulated by the DCC receptor and its downstream signaling cascade involving cAMP and Rac1 [44]. By assaying the DCC ligand, Netrin-1, with hybridization chain reaction (HCR), we surprisingly detected that *netrin1b* was enriched in cells surrounding the DRG neurons (Fig 3A and 3B). We also detected *netrin1b* in the spinal cord floor plate, consistent with its well-established role in axon guidance in the spinal cord (Fig 3C and 3D) [57,58]. To address if *netrin1b* is differentially expressed during regenerative period closure, we visualized *netrin1b* expression in *Tg(ngn1:GFP)* animals at 2 and 3 dpf. Quantitative analysis showed an increase in *netrin1b* expression dorsal to DRG neurons at 3 dpf compared to 2 dpf, coincident with the glial re-organization and regenerative period closure (Fig 3B). In contrast, the expression of *netrin1b* in the spinal floorplate was not different between 2 and 3 dpf animals (Fig 3C and 3D). To determine if *netrin1b* changes after injury, we also assayed its expression 10 hpi in animals injured at 3 dpf and 4 hpi in animals injured at 2 dpf; these times represent the respective midpoint of regeneration attempts (when axons are approximately 50% through the regenerative process) at these ages. This quantification revealed that elevated levels of *netrin1b* expression dorsal to the DRG were maintained 10 hpi in 3 dpf animals, but expression levels remained low 4 hpi in 2 dpf animals (Fig 3B). These data indicate higher *netrin1b* availability directly in the regenerative axon's path when the regenerative period is closed in 3 dpf animals. To confirm that glial cells were responsible for the robust *netrin1b* expression we observed around the DRG, we assayed its expression in *Tg(ngn1:GFP);Tg(sox10:nls-Eos)* animals after *sox10*+ nuclei surrounding one DRG were ablated in 3 dpf animals (Fig 3E). Prior to fixing the animals 4 hours post ablation (hpa), we confirmed glial cell death by the long-term loss of nuclear fluorescence and/or presentation of debris. HCR and quantitative analysis in these animals revealed a significant loss of *netrin1b* transcripts around DRG that underwent glial ablation compared to non-glia ablated DRG in the same animals (Fig 3F), indicating that glia are responsible for this robust expression.

To address if the expression of DCC, a receptor for Netrin, was changing during these ages, we performed HCR for *dcc* transcripts (Fig 3G). We fixed *Tg(ngn1:GFP)* animals at 2 and 3 dpf, as well as 10 hpi in 3 dpf animals. Quantification of *dcc* expression in DRG neurons revealed no statistical difference in *dcc* expression between 2 and 3 dpf animals (Fig 3H). However, in the injured context, particularly in the DRG that had undergone central axon axotomy, neuronal *dcc* expression was significantly increased 10 hpi in 3 dpf animals (Fig 3G–3H). These findings, coupled with our conclusion that glia dorsally migrate/re-organize and therefore increase the glial expression of *netrin1b* dorsal to the DRG, support a model in which both *dcc* and *netrin1b* are spatiotemporally organized in the tissue to close the regenerative period at 3 dpf.

One possible hypothesis to explain these results is that differential upregulation of *dcc* after injury closes the regenerative period. This hypothesis would predict that artificially increasing *dcc* expression would prematurely close the central axon's regenerative period. We tested this by overexpressing *dcc* in the DRG and assaying regeneration in animals injured at 2 dpf, when the regenerative period is normally still open. To do this, we injected *UAS:dcc-tdTomato* into *Tg(ngn1:GFP);Tg(sox10:Gal4 + myl7:GFP)* animals to enable *UAS*-driven overexpression [44,59]. We previously demonstrated this approach expresses a functional DCC tagged with tdTomato [44]. We specifically axotomized tdTomato+ DRG and assayed regeneration within 24 hpi (Fig 3I). Each of these DRG central axons regenerated within 24 hpi (n = 5), inconsistent with the hypothesis that DCC upregulation is sufficient to close the regenerative period (Fig 3J). Since *netrin1b*-expressing glia do not re-organize until 3 dpf, the premise of overexpressing *netrin* at 2 dpf to prematurely close the regenerative period is also inconsistent with such a hypothesis. These results led us to test the hypothesis that limiting DCC signaling could re-open the regenerative period in 3 dpf animals, after the regenerative period has closed and when *netrin1b*-expressing glia have re-organized dorsally (surrounding the regenerative axon).

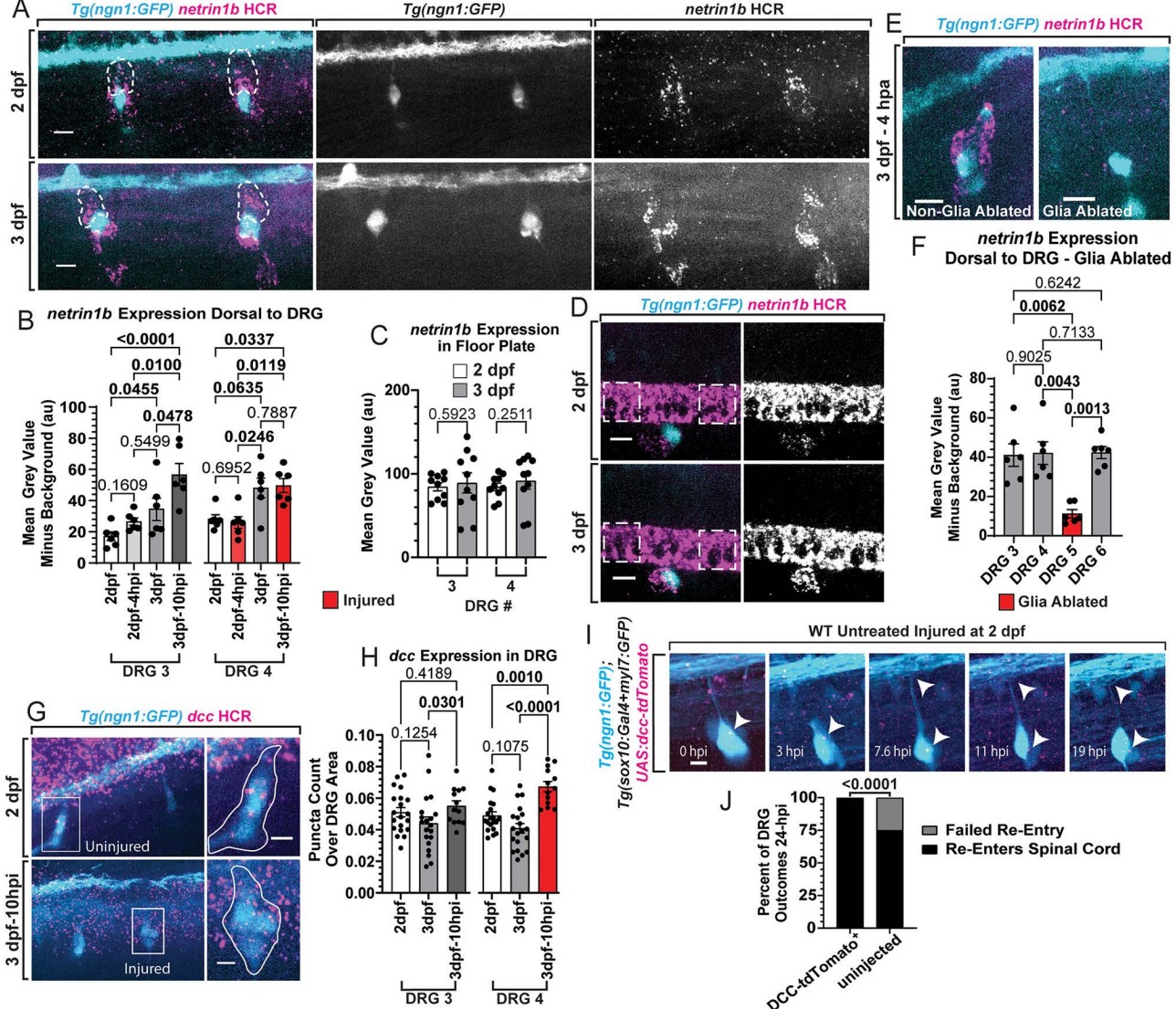

**Fig 3. Netrin-expressing glia re-organize and increases *netrin1b* dorsal to the DRG during closure of the regenerative period. A)** Max projection images of 2 and 3 dpf *Tg(ngn1:GFP)* animals after hybridization chain reaction (HCR) staining for *netrin1b* (scale bar = 10 μm). White dashed tracings indicate how the dorsal expression of *netrin1b* was quantified – traced from the top of the DRG to the DREZ signified by the dorsal longitudinal fasciculus. **B)** Quantifications of *netrin1b* expressed dorsal to the DRG in animals at 2 dpf, 2 dpf – 4 hpi, 3 dpf, and 3 dpf – 10 hpi (± SEM). The mean grey value of *netrin1b* puncta dorsal to the DRG minus background was compared between groups with Kruskal-Wallis One-Way ANOVA (n = 6 animals per group). Red bars on graph indicate DRG 4 was axotomized. (DRG number represents the position of the DRG from anterior to posterior. Anterior-most DRG is 1.) **C)** Quantification of the mean grey values of *netrin1b* in the spinal floor plate adjacent to each side of the DRG between 2 and 3 dpf animals (± SEM) (n = 6 animals per group). Mean grey values of floor plate expression was obtained with a 200 μm² box for each measurement. Comparisons were made with Kruskal-Wallis One-Way ANOVA. **D)** Representative z-cropped max projection images (scale bar = 10 μm) displaying the *netrin1b* expression in the spinal floor plates of 2 and 3 dpf animals. Dashed white boxes indicate where the floor plate expression was obtained with 200 μm² boxes. **E)** Images of *netrin1b* HCR in glia ablated and non-glia ablated DRG in 3 dpf animals 4 hours post ablation (hpa) (scale bar = 10 μm). **F)** Quantification of *netrin1b* expressed dorsal to DRG minus background signal (± SEM). Red bar indicates DRG 5 had glia ablated. Kruskal-Wallis One-Way ANOVA (n = 6). **G)** Representative images of *dcc* HCR staining in DRG of *Tg(ngn1:GFP)* animals at 2 dpf and an axotomized DRG at 3 dpf – 10 hpi (scale bars = 5 μm). **H)** Quantification of the number of *dcc* puncta in DRG neurons divided by DRG area (± SEM) in 2 dpf (n = 20), 3 dpf (n = 20), and 3 dpf – 10 hpi (n = 13) animals. Red bar indicates DRG 4 was axotomized. Kruskal-Wallis One-Way ANOVA. **I)** Representative images of 2 dpf Tg*(ngn1:GFP);Tg(sox10:gal4 + myl7:GFP)* animal injected with *UAS:dcc-tdTomato* pre-injury (0 hpi) as well as images throughout the timelapse (scale bar = 5 μm). White arrowheads indicate the DCC-tdTomato puncta in the DRG as well as the maintained position of the axon at the DREZ. **J)** Quantification of regeneration rates of DCC-tdTomato⁺ DRG (n = 5 animals, 1 DRG per animal) and uninjected animals injured at 2 dpf and assayed 24 hpi (n = 8 animals, 1 DRG per animal). Fisher's Exact test was used to compare re-entry and failed re-entry rates between groups. Raw data information can be found in S1 Data.

**Antagonizing DCC-cAMP-Rac1 signaling re-opens the regenerative period**

To determine if the DCC signaling axis is part of a genetic mechanism that closes the regenerative period of DRG central axons at 3 dpf, we targeted each step of the DCC-cAMP-Rac1 signaling process and scored regeneration. We hypothesized that antagonistic manipulations of this DCC signaling axis would enhance central axon regeneration because during development it modulates invadopodia that are essential for pioneer axon invasion [44]. To test this, we used *Tg(ngn1:GFP);Tg(gfap:NTR-mCherry)* animals to perform central axon axotomy on a single DRG per animal at 3 dpf and used genetic or molecular manipulations of each step of the DCC-cAMP-Rac1 signaling process for 24 hours after the injury (Fig 4A). At 24 hpi, the injured DRG central axon was re-imaged to assess regeneration into the spinal cord via orthogonal displacement quantifications (Figs 4C–4F and S1A1–S1F). In WT animals treated with only the vehicle control (1% DMSO), 0% of the central axons regenerated by 24 hpi (n = 10) (Fig 4B). However, in heterozygous *dcc* mutants (*dcc*$^{+/-}$) (ZM130198 [60]) bathed in 1% DMSO, 55% of axons regenerated back into the spinal cord by 24 hpi (n = 16) (p < 0.0001, Fisher's Exact test comparing re-entry and failed re-entry rates, Fig 4B). This ZM130198 allele incorporates a 5.2 kb retrotransposon insertion into the 5' UTR of *dcc* (106 bp upstream of the start codon) and was selected because it results in >90% reduction of *dcc* mRNA in homozygous mutant zebrafish [60]. Additionally, treatment at the time of injury with 25 μM rp-cAMP [61], an antagonistic cAMP analogue, or 1 μM NSC23766 [62], a Rac1 inhibitor, resulted in 50% (n = 10) and 55% (n = 20) of axons to regenerate into the CNS, respectively.

To determine if these molecules function in a similar signaling pathway to regulate the regenerative period, we performed double manipulations and determined genetic epistasis (Figs 4B–4F and S1A–S1F). Treatment with both rp-cAMP and NSC23766 resulted in 75% re-entry (n = 12), which was similar to either treatment alone (rp-cAMP vs rp-cAMP + NSC23766 p = 0.6020, Fisher's Exact test, Figs 4B and S1A–S1B). This supports the hypothesis that cAMP and Rac1 function in the same molecular pathway. Treating with 100 μM sp-cAMP [63], a cAMP agonist, resulted in 5.26% axon regeneration (n = 19) (Figs 4B and S1C–S1D). Co-treating with sp-cAMP and NSC23766 however, increased central axon regeneration to 57% (n = 14), indicative of Rac1 epistatically functioning downstream of cAMP. Combined manipulations of sp-cAMP, the cAMP agonist, in *dcc*$^{+/-}$ mutants reduced regeneration rates to 12.5% (n = 16) (Figs 4B and S1E–S1F), while co-treating with sp-cAMP and NSC23766 in *dcc*$^{+/-}$ animals improved regeneration back to 50% (n = 12) (p < 0.0001, Fisher's Exact test, Fig 4B–4F), indicating that the regenerative period is regulated by this signaling pathway in the order of DCC-cAMP-Rac1.

To test if such antagonistic manipulations of DCC-cAMP-Rac1 can re-open the regenerative period beyond 3 dpf, rather than simply extending it, we performed single central axon injuries in 5 dpf *Tg(ngn1:GFP);Tg(gfap:NTR-mCherry)* animals. These animals were either co-treated with rp-cAMP and NSC23766 or with 1% DMSO alone as a vehicle control for 24 hpi. This combination treatment was selected because it produced the highest regeneration rates at 3 dpf (Fig 4B). Additionally, it enabled us to add the pharmacological agents at 5 dpf, allowing the regenerative period to fully close before we tested if we could re-open it and promote regeneration. After 24 hpi, re-imaging and orthogonal displacement quantifications (Fig 4I–4J) confirmed that rp-cAMP and NSC23766 co-treatment (n = 10) resulted in 30% central axon re-entry compared to 0% in DMSO controls (n = 7) (Fig 4K). These findings are consistent with the idea that antagonistic manipulation of DCC-cAMP-Rac1 signaling can re-open the regenerative period.

One potential hypothesis for how these antagonistic manipulations are affecting the regenerative period is that they affect glial organization, which could influence regeneration independent of their effects on the regenerative axon. To test if antagonistic manipulations are affecting glial organization, we fixed WT uninjured 3 dpf (72 hpf) animals, as well as axotomized 3 dpf animals 12 hours after axotomy and treatment with 1% DMSO, 25 μM rp-cAMP, or 1 μM NSC23766. We also did this 12 hpi in *dcc*$^{+/-}$ animals treated with DMSO. We then performed immunohistochemistry (IHC) against Sox10 [64] to label glial nuclei surrounding the DRG. The x, y positions of the dorsal border of the DRG and the center of the dorsal-most Sox10 stained glial nuclei were obtained to quantify glial organization. The distance between the DRG and glial nuclei was calculated and plotted on target graphs (Figs 4L and S1G–S1I). These data demonstrated that neither *dcc*

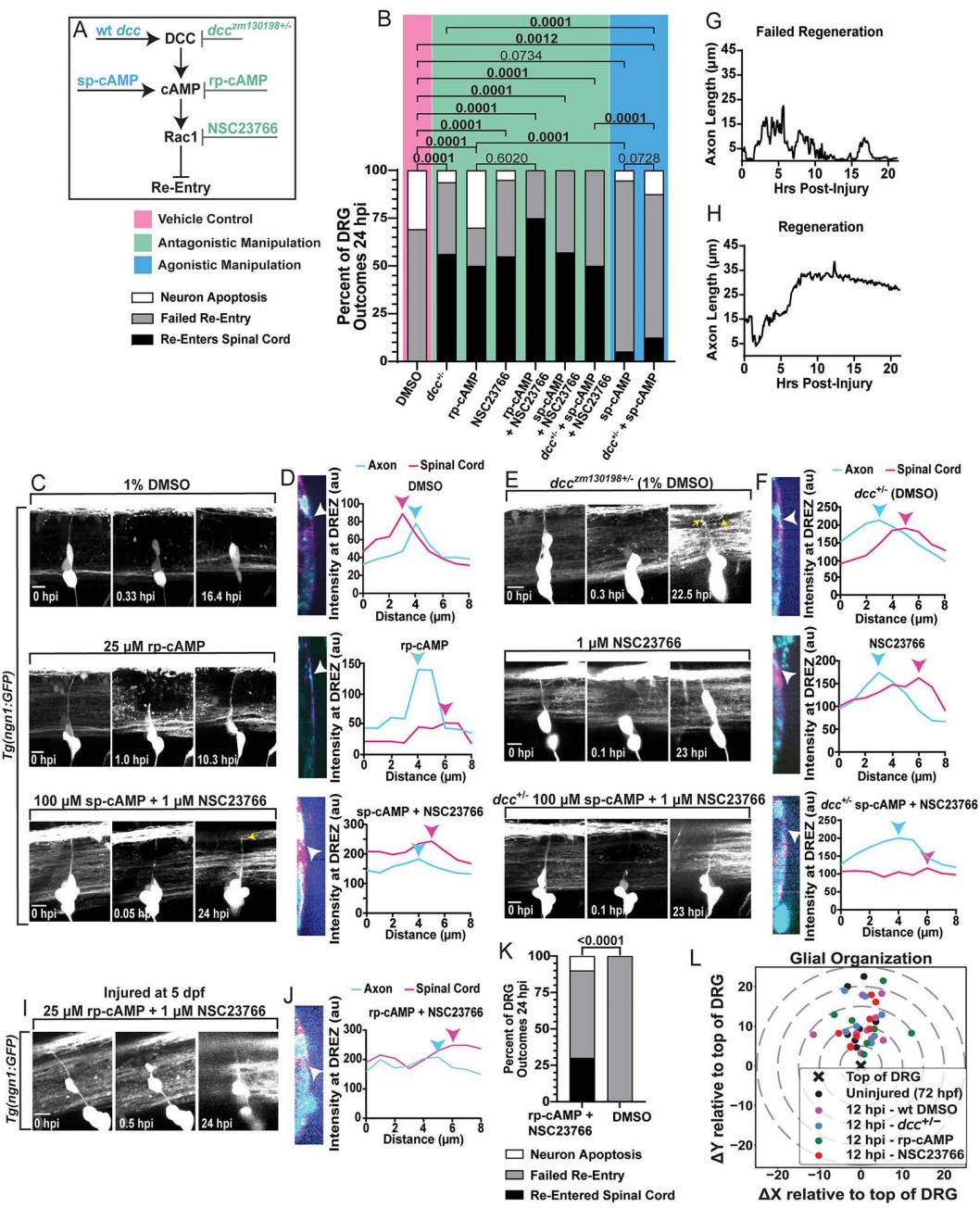

**Fig 4. Antagonistic manipulations of the DCC-cAMP-Rac1 signaling process re-opens the regenerative period. A)** Schematic of the agonistic (blue) and antagonistic (green) manipulations of each step of the DCC-cAMP-Rac1 signaling pathway. **B)** Quantifications of the percentage of outcomes 24 hpi under each manipulation. See key below 4A. Comparisons were made on the basis of central axon re-entry and failed re-entry using Fisher's Exact tests between treatment groups and DMSO vehicle controls. DMSO n = 10, *dcc*^+/− n = 16 (p < 0.0001); rp-cAMP n = 10 (p < 0.0001); NSC23766 n = 20 (p < 0.0001); rp-cAMP + NSC23766 n = 12 (p < 0.0001), sp-cAMP + NSC23766 n = 14 (p < 0.0001); *dcc*^+/− + sp-cAMP + NSC23766 n = 12 (p < 0.0001); sp-cAMP n = 19 (p = 0.0734); *dcc*^+/− + sp-cAMP n = 16 (p = 0.0012). **C and E)** Max projections of *Tg(ngn1:GFP)* animals pre-injury (0 hpi), post-injury (0.05-1.0 hpi) and 10-24 hpi in DMSO control and antagonistic treatment groups (scale bar = 10 μm). Treatment conditions for each are listed above each panel. Yellow arrows indicate bifurcated axons in panels where bifurcation is visible. **D and F)** DRG axons' orthogonal images and orthogonal displacement quantifications. White arrowheads indicate the axons' location in the orthogonal images. Cyan and pink arrowheads indicate the fluorescent peak of the axon and spinal cord boundary, respectively. **G)** Representative quantification of axon length (distance of growth cone from the DRG) during 22 hpi timelapses during failed (DMSO treated) and **H)** successful regeneration (rp-cAMP treated) attempts. **I)** Max projections of *Tg(ngn1:GFP)* 5 dpf animals pre-injury, post-injury (0.5 hpi) and 24 hpi treated with rp-cAMP + NSC23766 (scale bar = 10 μm). **J)** DRG axons' orthogonal image and orthogonal

displacement quantification. White arrowheads indicate where the fluorescent profile of the axon and spinal cord boundary were measured. Cyan and pink arrowheads on the graph indicate the fluorescent peak of the axon and spinal cord boundary, respectively. **K)** Quantification of the percentage of outcomes 24 hpi in rp-cAMP + NSC23766 (n = 10) versus DMSO (n = 7) control treated groups. Fisher's Exact test compared central axon re-entry and failed re-entry rates. **L)** Target graph of the x, y positions of the dorsal-most Sox10 IHC stained DRG glia relative to the dorsal border of the DRG (X at the center). WT Uninjured 3 dpf (72 hpf) animals (n = 8), 12 hpi WT DMSO treated animals (n = 10), 12 hpi $dcc^{+/-}$ DMSO treated animals (n = 7), 12 hpi rp-cAMP treated animals (n = 7), and 12 hpi NSC23766 treated animals (n = 7). The graph displays dashed (grey) lines in 5 µm increments. Raw data information can be found in S1 Data.

mutation nor treatment with rp-cAMP or NSC23766 affected glial re-organization compared to controls (S1I Fig), inconsistent with the hypothesis that the DCC-cAMP-Rac1 pathway modulates the regenerative period by altering glial organization. Taken together, these data indicate that this signaling pathway functions in the order of DCC-cAMP-Rac1 to close the regenerative period, and that antagonistically targeting this signaling pathway can re-open the period and promote regeneration.

## Antagonizing the DCC-cAMP-Rac1 signaling pathway increases invadopodia stability

Centrally-projecting DRG axons must navigate back to the DREZ to regenerate, during which filopodia are the predominant growth cone structure guiding its migration [44,65]. Once at the DREZ, invasion across the spinal cord boundary requires specialized actin-based growth cone structures called invadopodia, marked by robust actin accumulations and orthogonal basal protrusions [43–45,48,65]. The DCC-cAMP-Rac1 signaling pathway has been shown to spatiotemporally limit invadopodia formations during pioneer central axon development, until the pioneer axon growth cone reaches the DREZ and stable invadopodia assemblies are required for invasion into the CNS [44]. Based on our observations that glial re-organization and increased *netrin* presentation dorsal to the DRG coincides with regenerative period closure, we hypothesized that the regenerative period closes because DCC-cAMP-Rac1 signaling prevents the formation of stable invadopodia [42,65]. To determine the underlying mechanism by which DCC signaling closes the regenerative period, we manipulated each molecule in the pathway and assayed invadopodia. To visualize and quantify growth cone actin dynamics as a proxy for invadopodia formation and stability, we used *Tg(sox10:Gal4 + myl7:GFP);(UAS:LifeAct-GFP)* animals, in which LifeAct-GFP fluorescently labels filamentous actin in DRG cells, including the growth cone [66]. In 3 dpf animals, we axotomized the centrally-projecting axon of a single DRG per animal and then performed timelapse imaging (22–24 hpi, 5 min intervals) and measured the fluorescence intensity of actin at the center of the regenerative growth cone in each capture (S3–S7 Movies). We characterized invadopodia stability according to the duration of the fluorescent peaks in actin (above the average fluorescence, calculated over the entire timelapse). Peaks of 20–35 min were transient, unstable invadopodia, 40–55 min were intermediate invadopodia, and long duration peaks of 60–70 + mins were stable invadopodia (see key above Fig 5A).

We first confirmed that WT axons do not form stable invadopodia after injury in 3 dpf animals, as previously reported (DMSO vehicle controls, Fig 5A and S3 Movie) [42]. We then repeated the experiments while *dcc*, cAMP and Rac1 were manipulated (Fig 5A). Quantification of invadopodia revealed that reduction of DCC (in $dcc^{+/-}$ animals), antagonism of cAMP (rp-cAMP), and inhibition of Rac1 (NSC23766) all increased invadopodia stability compared to the appropriate controls (Fig 5A and S4–S6 Movies). Since we hypothesized that regeneration across the DREZ requires the formation of stable invadopodia, probing if these manipulations alter invadopodia stability (actin peak durations) specifically at this boundary is essential. Notably, the shift in stability from each manipulation was particularly evident when comparing only the actin peak durations produced when the growth cone was at the DREZ and stable invasive structures would be required for regeneration (Fig 5B). While DMSO controls (n = 8) averaged invadopodia with a duration of 41.04 ± 4.777 min at the DREZ, invadopodia in $dcc^{+/-}$ animals averaged 69.44 ± 9.020 min (n = 7) (Fig 5A–5B and S4 Movie). Further, rp-cAMP treatment, the cAMP antagonist, enabled invadopodia durations that averaged 76.58 ± 9.272 min (n = 6) at the DREZ (S5 Movie), while

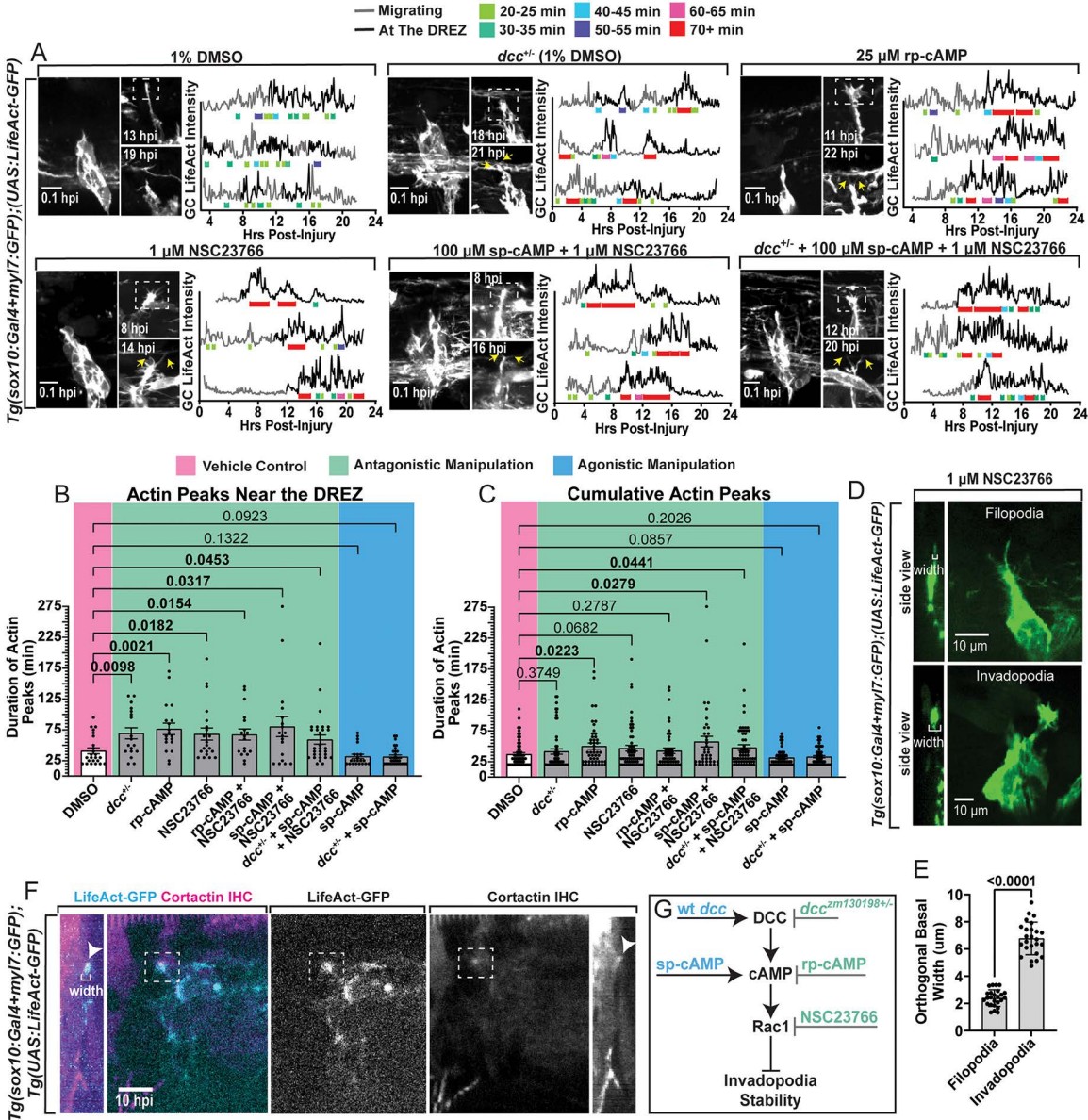

**Fig 5. Antagonistic manipulations of the DCC-cAMP-Rac1 cascade stabilizes invadopodia at the DREZ. A)** Max projections of *Tg(sox10:-Gal4 + myl7:GFP);(UAS:LifeAct-GFP)* DRG post-axotomy (0.1 hpi) and the regenerative growth cones during 24 hpi timelapses. White dashed boxes indicate the growth cone positioned at the DREZ, yellow arrows point to bifurcated axons. Treatment conditions for each are listed at the top of each panel. Quantifications of growth cone (GC) LifeAct intensity measurements in 3 representative growth cones under DMSO control and antagonistic manipulations over the 22-24 hpi timelapses. The grey portions of these line graphs indicate when the growth cone is navigating, and the black portions indicate when the growth cone was at the DREZ. Colored brackets under peaks of LifeAct fluorescence represent their duration. See key above. **B)** Quantifications of LifeAct peak durations only when the growth cone was at the DREZ (± SEM). **C)** Quantification of all LifeAct peak durations throughout the timelapses. In B & C, comparisons between groups were made with Brown-Forsythe and Welch ANOVA tests. **D)** Representative images of *Tg(sox-10:Gal4 + myl7:GFP);(UAS:LifeAct-GFP)* with orthogonal side views of the growth cones filopodia and invadopodia. White brackets in these side view images indicate orthogonal basal widths. **E)** Quantification of the orthogonal basal widths of filopodia and invadopodia (± SD) (25 growth cone measurements for each). Mann-Whitney T-test. **F)** Single z-plane images and orthogonal views of *Tg(sox10:Gal4 + myl7:GFP);(UAS:LifeAct-GFP)* growth cones 10 hpi treated with NSC23766 stained for Cortactin (IHC). **G)** Schematic of agonistic (blue) and antagonistic (green) manipulations at each step of the DCC-cAMP-Rac1 pathway regulating invadopodia stability. All scale bars = 10 μm. Raw data information for this figure can be found in S1 Data.

the cAMP agonist, sp-cAMP, averaged 31.90±2.544 min (n=7) (Figs 5A–5B and S2A). Moreover, in NSC23766, the Rac1 inhibitor, invadopodia averaged 68.33±9.804 min (n=7) in duration at the DREZ (Fig 5A–5B and S6 Movie). To determine if these manipulations regulated invadopodia globally rather than at the precise spatial location of invasion (DREZ), we displayed the distribution and frequency of all actin peak durations and compared invadopodia durations throughout the entire navigation process, including at the DREZ, between groups (Fig 5C). These results demonstrated that the stabilizing effects on invadopodia were present at the DREZ (Fig 5B) and not throughout the entire navigation process (Fig 5C). In contrast to the antagonistic manipulations of the pathway, agonistic manipulations of DCC signaling recapitulated the WT phenotype and produced primarily transient invadopodia formations (Figs 5A–5C and S2A).

To test if these molecules function in a common signaling process, we again performed combination treatments for epistasis analysis while measuring invadopodia during regeneration. Co-treatment with sp-cAMP and NSC23766 in WT animals resulted in invadopodia durations averaging 80.28±16.30 min (n=5) at the DREZ, indicating Rac1 functions downstream of cAMP in regulating invadopodia stability (Fig 5A–5B). Similar to sp-cAMP alone (S7 Movie), sp-cAMP treatment in $dcc^{+/-}$ animals reduced invadopodia durations at the DREZ, averaging 31.55±2.708 min (n=7), indicating cAMP functions downstream of DCC (Figs 5B and S2A); however, co-treating with both sp-cAMP and NSC23766 in $dcc^{+/-}$ animals increased invadopodia durations at the DREZ to 59.17±7.413 min (n=7) (Fig 5A–5B). Together, these findings indicate that invadopodia stability is regulated by this signaling pathway in the order of DCC-cAMP-Rac1 during regeneration, corresponding with our re-entry analyses.

In timelapse movies, we could detect the key morphological differences between a filopodia-based growth cone and an invadopodia-based growth cone (Fig 5D) [42–44,65]. During invadopodia formations, the robust, long duration, actin accumulations at the center of the regenerative growth cone were accompanied by orthogonal basal protrusions toward the DREZ, which measured significantly wider in orthogonal width measurements compared to filopodia (Fig 5D–5E) (n=25 growth cone measurements for both filopodia and invadopodia across 5 biological replicates). While previous work has shown that long duration LifeAct-GFP peaks with basal-projecting structures are indicative of invadopodia assembly, we used a complementary approach to identify mature invadopodia [43,44,46]. Cortactin is enriched at invadopodia in growth cones and cancer cells, acting as a key regulator of invadopodia assembly and maturation, with important functions in scaffolding and signaling [67,68]. We therefore axotomized a DRG central axon of 3 dpf *Tg(sox10:Gal4 + myl7:GFP);(UAS:LifeAct-GFP)* animals and fixed them 10 hpi to perform IHC against Cortactin. Prior to fixing, these animals were treated with the Rac1 inhibitor (NSC23766), as this treatment reliably produced stable invadopodia and maximized the likelihood of observing invadopodia post fixation. From this, we observed Cortactin localized to invadopodia-like structures in the growth cone (Fig 5F). Taken together, these data support the hypothesis that DCC signaling closes the regenerative period by actively reducing the stability of invadopodia formations (Fig 5G).

## Cell-specific manipulation of DCC and Rac1 alter invadopodia stability

Based on HCR expression analysis of *dcc* and *netrin*, we next hypothesized that DCC signaling functions autonomously in DRG. To test this, we first manipulated DCC cell-specifically using the functional DCC construct, *UAS:dcc-tdTomato*, injected at the one-cell stage of *Tg(sox10:Gal4 + myl7:GFP);(UAS:LifeAct-GFP); dcc^{+/-}* embryos to increase expression of *dcc* specifically in DRG cells [44]. Since a specific driver that expresses reliably in DRG neurons vs glia has not been identified in zebrafish, this experimental paradigm tested the specific requirement in DRG cells. To provide more specificity, we selected mosaic animals that express tdTomato in neurons but we could not completely rule out glial expression. The $dcc^{+/-}$ animals exhibit increased invadopodia stability and regenerative outcomes, so we predicted that increasing *dcc* expression would reduce invadopodia stability and regenerative potential to WT levels compared to uninjected $dcc^{+/-}$ animals. At 3 dpf, the central axon of a DCC-tdTomato⁺ DRG was axotomized and then time lapsed for 24 hours to record and measure invadopodia stability (S8–S10 Movies). We then scored if these regenerative axons either maintained their position at the DREZ or subsequently retracted to discern if regenerative outcomes were correspondingly affected.

We first confirmed that we could detect tdTomato puncta in a DRG of injected animals prior to injury, post-injury and throughout the timelapse (Fig 6A–6C). In the $dcc^{+/-}$ animals, DRG expressing DCC-tdTomato displayed reduced invado-podia stability (S8 Movie), averaging 31.46 ± 2.472 min (n = 4) at the DREZ, recapitulating invadopodia durations seen in injected WT animals (31.90 ± 2.372 min, n = 4, S9 Movie) and uninjected WT animals (29.35 ± 2.058 min, n = 4) (Fig 6A, 6B and 6D). These results were distinct from uninjected $dcc^{+/-}$ animals without DCC supplementation when comparing invadopodia durations at the DREZ (averaging 57.00 ± 8.162 mins at the DREZ, n = 3, Fig 6D), but this distinction was lost when compared cumulatively (S3A Fig). These data demonstrate that DCC functions, specifically in DRG cells, to desta-bilize invadopodia. While the cell autonomy of cAMP could not be determined, we tested if manipulating cAMP during this cell-specific manipulation of DCC altered invadopodia. To do this we treated $dcc^{+/-}$ animals expressing DCC-tdTomato with rp-cAMP, the cAMP antagonist (Fig 6C and S10 Movie). This treatment confirmed that rp-cAMP is capable of reducing downstream signaling of DCC, increasing invadopodia durations to 54.07 ± 7.478 min (n = 5) at the DREZ (Fig 6D).

To next determine if DRG-specific manipulations also altered the regenerative capacity, we scored if these regenerative axons either maintained their position at the DREZ or subsequently retracted by the end of the 24 hour timelapse. For technical reasons because of fluorescent reporter conflict (e.g., DCC-tdTomato vs *gfap:mCherry*), we could not perform orthogonal displacement quantifications with the glial limitans. While an axon maintaining its position at the DREZ 24 hpi is indicative of regeneration, axonal retraction is a direct indicator of regenerative failure. Therefore, the proportions of these outcomes were compared between groups to quantify regenerative potential. These measurements showed that untreated animals with DCC-tdTomato-expressing DRG had significantly more axons that subsequently retracted (Fig 6E). Untreated $dcc^{+/-}$ animals (n = 7) only had 28.57% of axons remain at the DREZ, and WT siblings (n = 6) had 50% remain (Fig 6E). These findings are particularly impactful given our previous demonstration that *dcc* mutation reduces central axon regenerative failure. In contrast, $dcc^{+/-}$ animals treated with rp-cAMP (n = 5) had 80% of their axons remain at the DREZ (Fig 6E). These results are consistent with the idea that DCC functions, specifically in the DRG cells, to close the regenerative period by destabilizing invadopodia.

To test if Rac1 activity, specifically in the DRG, controls the regenerative period, we injected a photoactivatable Rac1 construct *(UAS:pa-Rac1-mCherry)* into *Tg(sox10:Gal4 + myl7:GFP);(UAS:LifeAct-GFP)* embryos, allowing us to again manipulate the signaling cascade in DRG cells [69]. As we previously published, pa-Rac1 is activated by 445 nm light [43,44,69]. Therefore, uninjected animals, as well as injected animals without 445 nm light exposure, could be used as controls. Since our previous experiments showed that rp-cAMP stabilizes invadopodia dynamics during regeneration and that Rac1 functions downstream of cAMP in the signaling process, we hypothesized that pa-Rac1-mCherry+ DRG exposed to 445 nm light would reverse rp-cAMP-induced invadopodia stability back to WT/untreated levels. We also predicted that photoactivated Rac1 would correspondingly reduce regenerative potential. Uninjected animals treated with rp-cAMP and exposed to 445 nm light (every 5 min) maintained the ability to form long-duration, stable, invadopodia at the DREZ (averaging 55.54 ± 6.162 min, n = 6) (Fig 6H and 6I), which was comparable to the pa-Rac1-mCherry+ DRG treated with rp-cAMP unexposed to 445 nm light (averaging 49.26 ± 4.284 min at the DREZ, n = 6) (Fig 6F and 6I and S11 Movie). However, pa-Rac1-mCherry+ DRG treated with rp-cAMP that were exposed to 445 nm light displayed destabilized inva-dopodia dynamics (averaging 33.61 ± 2.463 mins at the DREZ, n = 7, Fig 6G and 6I and S12 Movie), which was distinct from the durations quantified at the DREZ in the unexposed group (Fig 6F-6I). These exposed and unexposed pa-Rac1-mCherry+ DRG were not significantly different on a cumulative basis (S3B Fig).

To test the cell-autonomous effects of Rac1 activity on regenerative potential, we again scored the proportions of regen-erative axons that either maintained its position at the DREZ or retracted by the end of the 24 hpi timelapse (Fig 6J). All animals in this analysis were treated with rp-cAMP. We predicted that photoactivation of pa-Rac1 (exposure to 445 nm light) would negatively affect regenerative potential compared to unexposed animals and uninjected animals. Here, we observed that only the animals/DRG that had pa-Rac1 photoactivated had significantly higher retraction rates (57.14%, n = 7, Fig 6J). This was in stark contrast with unexposed/unactivated pa-Rac1+ DRG (n = 6), in which only 16.67% of their axons

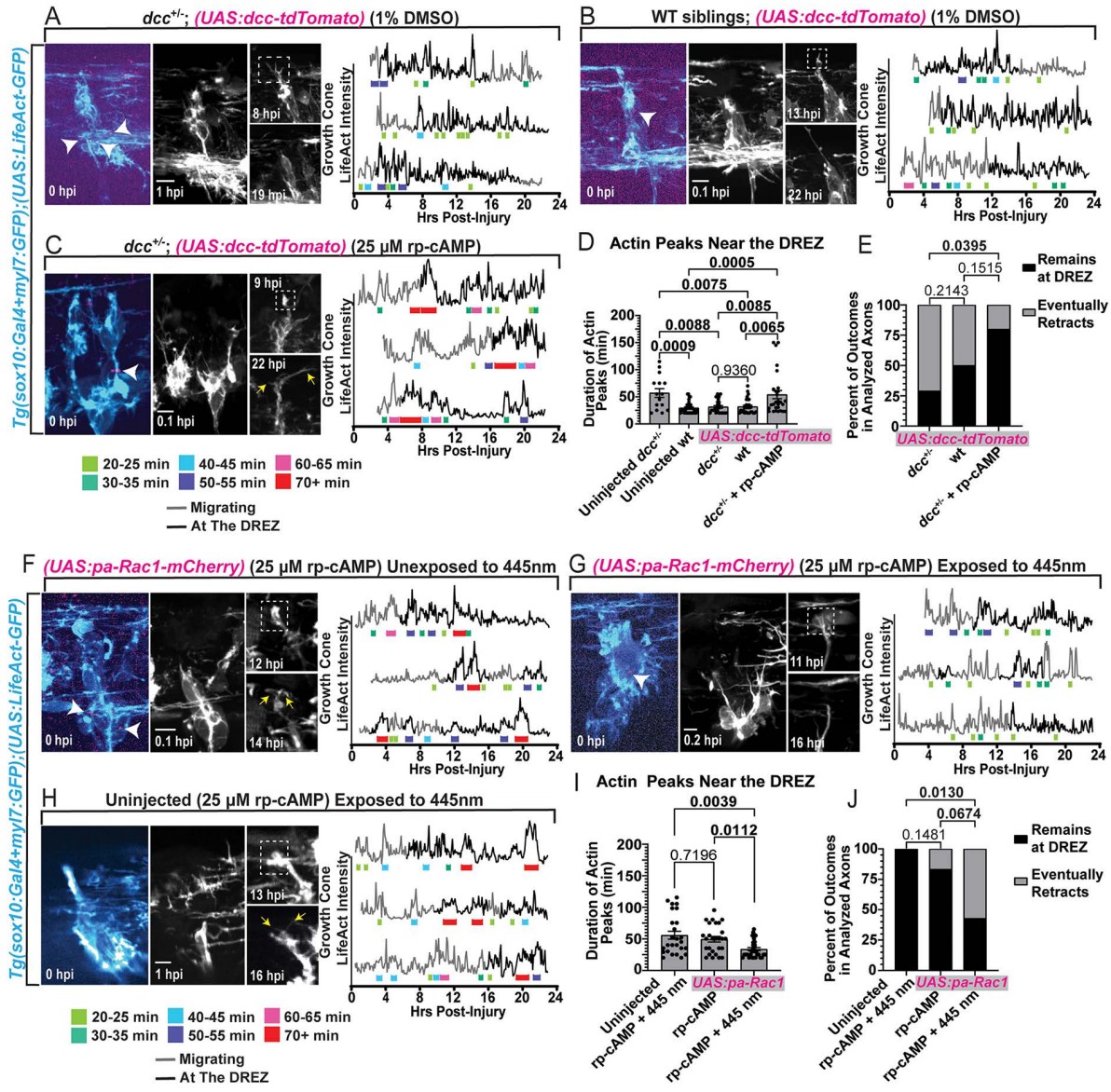

**Fig 6. Cell-specific manipulations of DCC and Rac1 indicate cell-autonomous regulation of invadopodia stability. A-C)** Max projections of DCC-tdTomato⁺ puncta (white arrowheads) in *Tg(sox10:Gal4+myl7:GFP);(UAS:LifeAct-GFP)* DRG pre-injury, the DRG post-injury, and images of the growth cone during 24 hpi timelapses in **(A)** dcc⁺/⁻ animals, **(B)** WT siblings and **(C)** *dcc⁺/⁻* animals treated with rp-cAMP. The genotype and treatment condition are listed above each panel. The white dashed boxes indicate the growth cone positioned at the DREZ. Yellow arrows point to bifurcated axons. Quantification of growth cone LifeAct intensity measurements in 3 growth cones under each condition. The grey portion of these line graphs indicate that the growth cone is navigating, and black portions indicate it is at the DREZ. Colored brackets beneath LifeAct fluorescent peaks indicate their duration (see key below 6C). **D)** Quantification of LifeAct peak durations only when the growth cone was at the DREZ (± SEM). Kruskal-Wallis One-Way ANOVA test. **E)** Quantification of the proportion of regenerative axons that remained at the DREZ or retracted by the end of the 24 hpi timelapse between DCC-tdTomato⁺ DRG of *dcc⁺/⁻* (n=7), wildtype (n=6) and *dcc⁺/⁻* animals treated with rp-cAMP (n=5). Chi-Squared tests. **F-H)** Max projections of pa-Rac1-mCherry⁺ puncta in *Tg(sox10:Gal4+myl7:GFP);(UAS:LifeAct-GFP)* DRG (white arrow heads) pre-injury, the DRG post-injury, and images of the growth cone during 24 hpi timelapses in injected and **(H)** uninjected animals treated with rp-cAMP. The injection status and exposure conditions to the activating 445 nm light are listed above each panel. The white dashed boxes indicate the growth cone positioned at the DREZ. Yellow arrows point to bifurcated axons. Quantification of growth cone LifeAct intensity measurements in 3 growth cones under each condition. The grey portion of these line graphs indicate when the growth cone is navigating, and black portions indicate when it is at the DREZ. Colored brackets under the LifeAct fluorescent peaks indicate their duration (see key below 6B). **I)** Quantifications of the actin peak durations while growth cones were at the DREZ (± SEM). Kruskal-Wallis One-Way ANOVA test. **J)** Quantification of the proportions of regenerative axons that remained at the DREZ or retracted by the end of the 24 hpi timelapse between pa-Rac1-mCherry⁺ DRG of rp-cAMP treated animals that were exposed (n=7) or were not (n=6) exposed to 445 nm light and uninjected animals treated with rp-cAMP and exposed to 445 nm light (n=6). Chi-Squared tests. Raw data information for this figure can be found in S1 Data.

retract (83.33% remained at the DREZ, Fig 6J). Furthermore, uninjected animals (n = 6) that were exposed to 445 nm light had 100% of their axons remain at the DREZ (Fig 6J). This indicates that 445 nm light exposure does not inherently affect regenerative axons, but photoactivation of pa-Rac1 does. Collectively, these data support the model that Rac1 functions autonomously in DRG cells to reduce the stability of invadopodia, which correspondingly hampers regenerative potential.

## Opening the regenerative period by antagonizing DCC signaling restores somatosensory circuits

Regenerative period closure prevents the functional recovery of somatosensory circuits between the DRG and spinal cord after injury. To determine if re-opening the regenerative period with manipulations of DCC signaling restores somatosensory circuits, we utilized the activity of a genetically encoded calcium indicator expressed in the DRG and spinal neurons, GCaMP6s, as a proxy for neuronal activity upon exposure to a sensory stimulus. DRG neurons exhibit $Ca^{2+}$ transients that are detected by increases in GCaMP6s fluorescence [70–72]. To assess the restoration of the somatosensory circuit, we determined if spinal neurons exhibited GCaMP6s transients in a synchronous manner with DRG neurons upon a cold-water stimulus (4°C) (S13 Movie) [71]. We hypothesized that the antagonistic manipulations of DCC signaling that re-opens the regenerative period would also enable the re-establishment of somatosensory circuits. In such a case, synchronized neuronal activity between DRG and spinal neurons would be observed. This assay is limited to the temporal resolution of GCaMP6s dynamics, which fluoresces quickly upon neuronal activation and has a delayed decay of fluorescence intensity [73]. We observed that an absence of synchrony between DRG neurons and spinal neuron transients was exhibited by a considerable delay of neuronal activation (~2–3 seconds) in the spinal neuron population. Z-scores above 2.0 were used to identify active neurons [72].

To determine if somatosensory-induced behaviors are being restored from our manipulations of the DCC signaling pathway, we measured circuit function in animals that had 8 consecutive DRG axons axotomized (DRG #4–11). This published assay creates a discernable difference in circuit and behavior dynamics after exposure to sensory stimuli [42,74]. We first performed series axotomies to 8 DRG (DRG #4–11) in 3 dpf *Tg(NeuroD:Gal4 + myl7-GFP);(UAS:GCaMP6s)* and *Tg(sox10:mRFP)* animals [46,75,76]. Animals were treated for 24 hours and then recovered untreated for an additional 24 hours (Fig 7A–7B). At 48 hpi, we performed the $Ca^{2+}$ imaging assay with an evoked cold-water stimulus to assess the synchrony of previously axotomized DRG and spinal neuron activity as an indication of functionally restored sensory circuits (Fig 7C). All animals included in this assay displayed at least one DRG neuron per ganglia that was rapidly active upon cold water exposure (Fig 7D). In uninjured animals, an average of 97.78 ± 2.222% (n = 7) of spinal neurons displayed activity that occurred rapidly, synchronized with the activity of the DRG (Fig 7E and S13 Movie). However, the injured DMSO vehicle treated and untreated control animals exhibited a significant reduction of this synchrony, averaging 71.90 ± 8.724% (n = 7) and 64.24 ± 11.69% (n = 7) of active spinal neurons in synchrony with the DRG, respectively (Fig 7E and S14 Movie). Similarly, injured animals treated with the cAMP agonist, sp-cAMP, averaged 68.94 ± 11.370% (n = 5) of spinal neuron activity synchronized with the DRG (Fig 7E). This was in stark contrast to the antagonistically manipulated groups (Fig 7E). In *dcc*[+/−] animals, 89.41 ± 3.293% (n = 6) of spinal neurons were synchronized with the DRG (Fig 7E and S15 Movie). In rp-cAMP and NSC23766 treated animals, 92.67 ± 4.760% (n = 5) and 93.59 ± 2.722% (n = 5) of spinal neurons were synchronized with the previously injured DRG, respectively (Fig 7E and S16 Movie). While the number of DRG neurons responding per DRG was variable and dependent on GCaMP6s expression, no difference was detected in the propensity for spinal neurons to exhibit synchronized activity between cases of one DRG neuron versus multiple neurons being activated. Together this data is consistent with the idea that re-opening the regenerative period of centrally-projecting DRG axons enables the restoration of somatosensory circuits.

## Opening the regenerative period recovers somatosensory-induced behaviors

Functional recovery of somatosensory circuits after injury would also be expected to restore behavioral responses [42]. When exposed to cold (4°C) water, larval animals display a shivering behavior consisting of head and tail movements

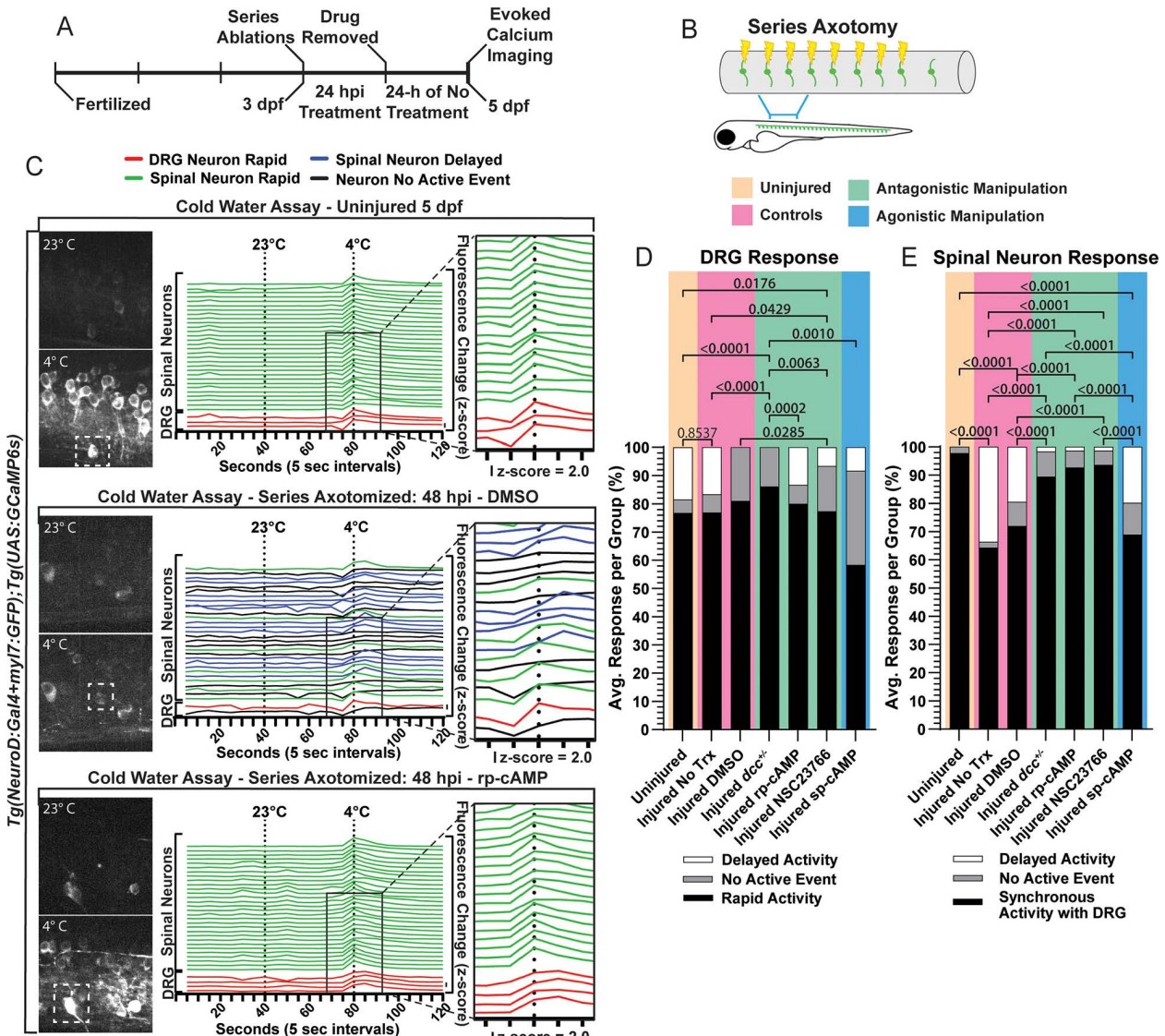

**Fig 7. Antagonistically manipulating DCC signaling promotes the recovery of synchronous activity between previously axotomized DRG neurons and spinal neurons. A-B)** Timeline of the sensory circuit assay, wherein following series axotomy **(B)** the animals were treated with manipulations of the DCC-cAMP-Rac1 pathway for 24 hpi and then allowed to recover for another 24 hrs. **C)** Representative max projection images and quantification of the GCaMP6s calcium indicator transients after room temperature (23°C) and cold (4°C) water additions. White dashed boxes indicate the DRG. Individual traces (lines) represent GCaMP6s transients in a single neuronal soma over 120 seconds. DRG neurons are plotted at the bottom of the graph and spinal neurons above. The color of the trace indicates the characterized activity based on z-score change (see key below 7A). The vertical line in front of "z-score = 2.0" indicates the vertical increase in the individual traces (lines) needed to reach a z-score of 2.0 and be considered an activated neuron. **D)** Quantification of the activity response of DRG neurons per animal in each treatment group. Comparisons were made based on the proportions of Delayed Activity and Rapid Activity between groups with Fisher's Exact tests. **E)** Quantification of spinal neuron responses in each treatment group immediately following cold water exposure. Comparisons were made with Fisher's Exact tests based on the proportions of Delayed Activity and Synchronous Activity with the DRG between groups. For 7D and 7E Uninjured animals n = 7, injured No Treatment (Trx) n = 7, injured DMSO n = 7, injured $dcc^{+/-}$ n = 6, injured rp-cAMP n = 5, injured NSC23766 n = 5, injured sp-cAMP n = 5. In 7E, Fisher's Exact tests were used to compare the proportions of Delayed versus Synchronous spinal neuron activity, and when compared to Uninjured animals: injured No Trx p < 0.0001; injured DMSO p < 0.0001; injured $dcc^{+/-}$ p = 0.2305, injured rp-cAMP p = 0.4895; injured NSC23766 p = 0.4922; injured sp-cAMP p < 0.0001. Raw data information for this figure can be found in S1 Data.

without forward locomotion. This behavior depends on intact DRG circuits in zebrafish both in regenerative and developmental contexts [42,44]. At 3 dpf, *Tg(ngn1:GFP);Tg(gfap:NTR-mCherry)* animals underwent the series axotomy and 24 hpi treatments to manipulate DCC-cAMP-Rac1 (Fig 8A-8B). After 24 hpi treatment, the animals were allowed to recover for another 24 hours with no additional treatment (Fig 8A). After 48 hpi, DRG #4–11 were re-imaged in each animal to perform orthogonal displacement quantifications and determine how many of the 8 injured central axons regenerated (Fig 8D), and then animals were subjected to the behavioral assay. In the assay, animals were first immersed in room temperature water (23°C) and then cold water (4°C), and 600 frame movies were collected (50 ms intervals) in each water condition. To compare groups, the number of frames in which animals displayed the shivering behavior (described above) were measured. We hypothesized that antagonistic manipulation of DCC signaling would re-open the regenerative period and enable functional recovery of somatosensory circuits, causing the animals to display longer shivering times. Uninjured controls displayed shivering in ~50% of the frames (tan region of Fig 8C and 8E). However, at 48 hpi, injured DMSO treated controls averaged 28.18±4.697% (n = 11) of frames spent shivering (pink region of Figs 8C and S4). In these animals, 25±2.919% of axotomized DRG axons had regenerated (pink region of Fig 8D). Similarly, agonistic treatments of sp-cAMP, sp-cAMP in $dcc^{+/-}$ animals, and DMSO control treatment in WT siblings had central axon regeneration rates of 29.61±2.901%, 17.86±2.845%, and 28.57±4.024%, respectively (blue region of Fig 8D). Correspondingly, these animals had reduced shivering times, shivering in 33.63±3.659% (n = 19), 35.11±4.331% (n = 14), and 36.68±6.266% (n = 11) of frames, respectively (blue region of Figs 8C and S4).

These results are in stark contrast with the antagonistic manipulations (green regions of Figs 8C, 8D, and S4). The $dcc^{+/-}$ animals had central axon regeneration rates of 45.45±3.728% and shivered 49.83±4.513% (n = 20) of the time (Fig 8C–8D). Furthermore, rp-cAMP treatment alone, as well as rp-cAMP in $dcc^{+/-}$ animals, resulted in regeneration rates of 53.33±5.652% and 50.78±3.864%, respectively, and shivered 52.47±4.593% (n = 15) and 59.20±6.879% (n = 16) of the frames, respectively (Fig 8C–8D). Rac1 inhibition with NSC23766 alone, as well as in co-treatment with sp-cAMP resulted in regeneration rates of 50.83±4.307% and 55.00±3.819%, respectively (Fig 8D). These animals also displayed longer shivering times, 59.00±6.489% (n = 15) and 44.40±3.752% (n = 15) of frames, respectively (Fig 8C). While many of the animals spent much of their time in the 23°C water stationary, all animals were of good health and were capable of swimming. In animals that were stationary in the 23°C, a nose touch was used to elicit an escape response to confirm the capability of swimming. Analysis of WT and $dcc^{+/-}$ animals 48 hpi between water conditions demonstrate that the shivering phenotype is a distinct response that is not caused by an inability to swim (Fig 8E–8G). Taken together, these results indicate that antagonistic manipulations of DCC-cAMP-Rac1 signaling re-opens the regenerative period and enhances DRG central axon regeneration. These manipulations correspondingly recover somatosensory-induced behaviors, indicative of improved functional regeneration of sensory circuits between previously axotomized DRG and the CNS.

## Discussion

Regenerative period closure is an important component of development, securing tissue architecture for proper function and protecting organisms from aberrant cellular growth. However, in injured and diseased states, this leaves tissues with diminished plasticity and regenerative capacity, which can result in permanent or prolonged dysfunction [4,6,7]. The centrally-projecting axons of DRG neurons experience a significant decline in regenerative capacity shortly after their development, seen across many vertebrate species, including humans, where such injuries are devastating to infants, children, and adults [26,27,29,77]. Our model demonstrates that the regenerative period closes between 2 and 3 dpf in zebrafish. This precise and narrow regenerative period provides the field with an impactful model system with clinical relevance. Data from our *in vivo* model suggests that all the relevant regenerative molecular machinery is still present during this closure but the formations of stable invadopodia required for successful axon regeneration are dysregulated by a Netrin-mediated DCC signaling axis, brought on by a glial re-organization around the DRG (Fig 9). Together these data support the idea that closure of the regenerative period depends on an active suppression of regenerative machinery that is distinct from development of the DRG central axon.

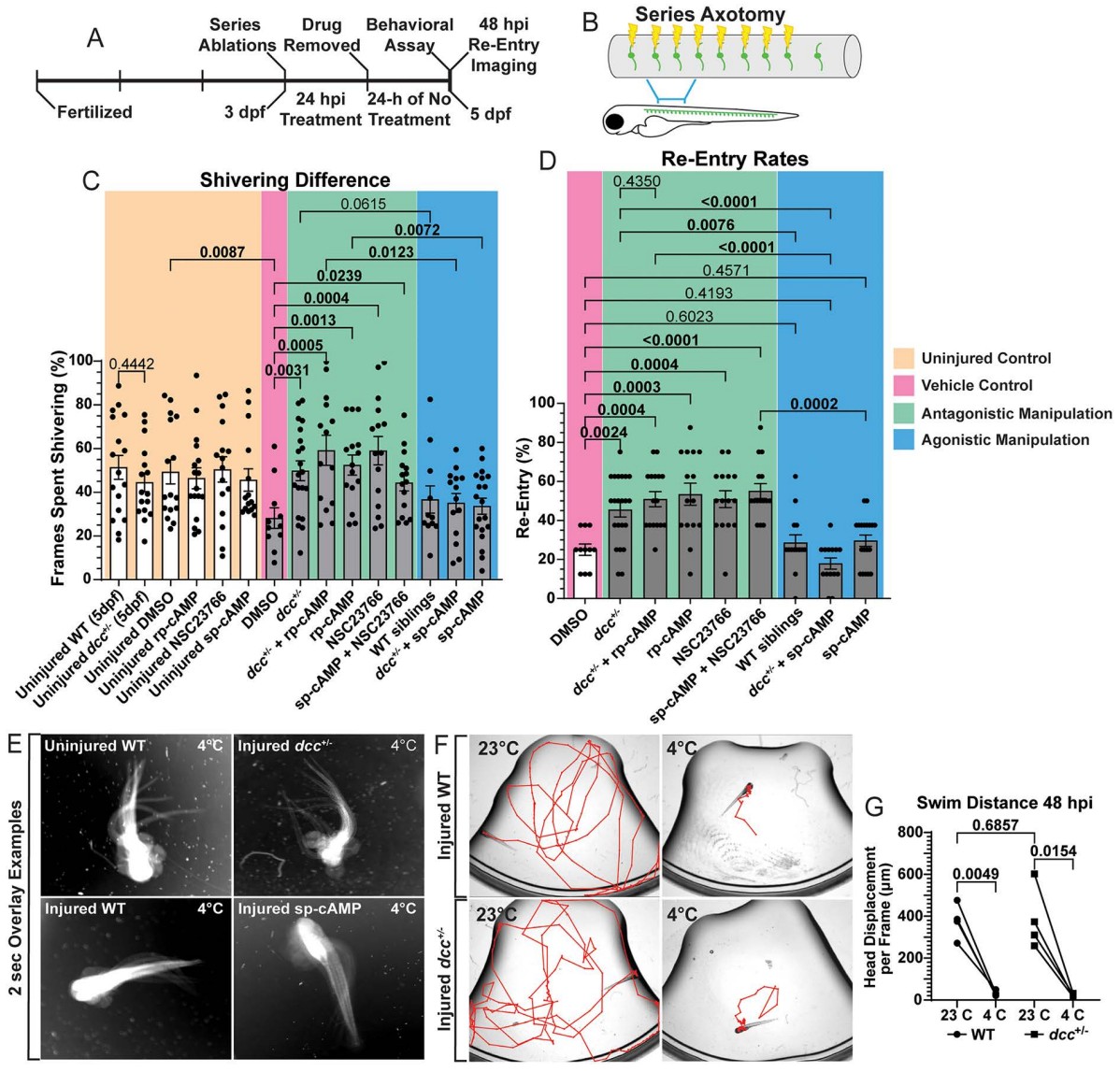

**Fig 8. Antagonistically manipulating DCC signaling enables the recovery of somatosensory-induced behaviors. A-B)** A timeline of the behavioral assay, wherein following series axotomy **(B)** the animals were treated for 24 hpi to manipulate the DCC-cAMP-Rac1 signaling pathway then allowed to recover for 24 hours. At 48 hpi DRG were re-imaged and then animals underwent the behavioral assay. **C)** Quantification of the average percentage of time spent shivering in uninjured and injured animals under each manipulation (± SEM). Kruskal-Wallis One-Way ANOVA. **D)** Quantification of central axon regeneration rates per animal (± SEM). Kruskal-Wallis One-Way ANOVA. For 8C and 8D Uninjured WT n = 17, Uninjured *dcc*<sup>+/−</sup> n = 16, Uninjured DMSO n = 16, Uninjured rp-cAMP n = 17, Uninjured NSC23766 n = 16, Uninjured sp-cAMP n = 15, DMSO n = 11, *dcc*<sup>+/−</sup> n = 20, *dcc*<sup>+/−</sup> + rp-cAMP n = 14, rp-cAMP n = 15, NSC23766 n = 15, sp-cAMP + NSC23766 n = 15, WT siblings n = 11, *dcc*<sup>+/−</sup> + sp-cAMP n = 14, sp-cAMP n = 19. **E)** 2-sec overlay examples of the shivering phenotype in uninjured WT, injured WT and injured *dcc*<sup>+/−</sup> animals (48 hpi) in cold (4°C) water. **F)** Representative traces (red lines) of animal locations (tracked by head displacement) 48 hpi in 23°C and 4°C water. **G)** Quantification of head displacement between the two water conditions and genotypes. Mann-Whitney T-test compares swimming distances between WT and *dcc*<sup>+/−</sup> animals in 23°C water 48 hpi (p = 0.6857). Paired T-tests compares head displacement per frame between the two water conditions in WT animals (p = 0.0049) and *dcc*<sup>+/−</sup> animals (p = 0.0154) (n = 4 in each group). Raw data information can be found in S1 Data.

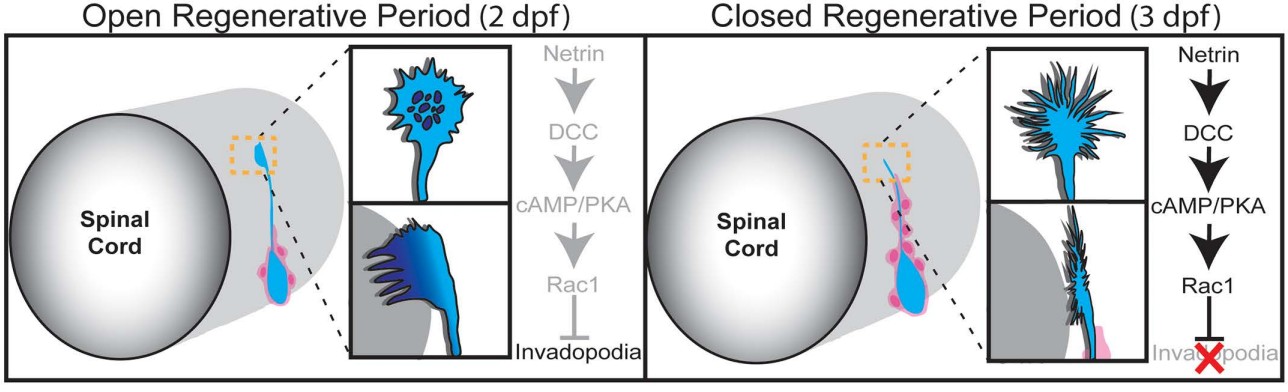

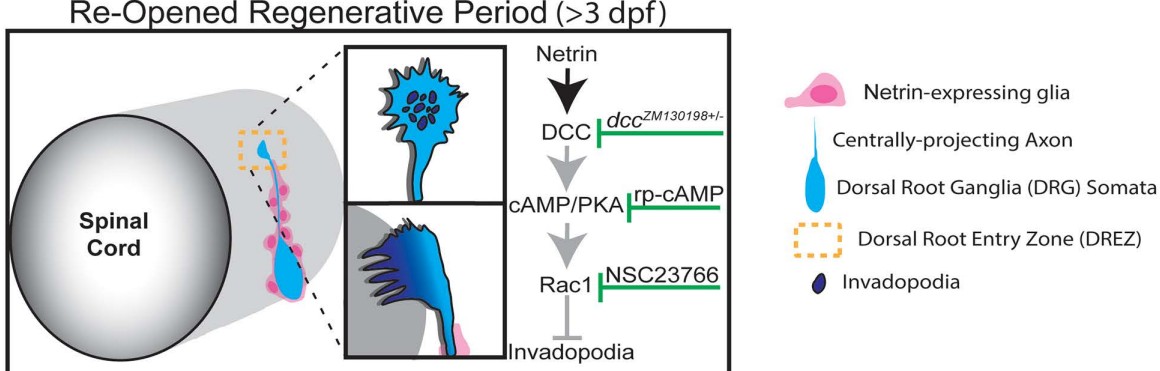

**Fig 9. Graphical Abstract illustrating the proposed mechanism regulating the regenerative period for centrally-projecting DRG axons (blue).** At 2 dpf, when the regenerative period is open, Netrin-expressing glia (pink) have not yet re-organized dorsally. As a result, Netrin-mediated DCC signaling is inactive (greyed out) as the regenerative growth cone approaches the DREZ, permitting stable invadopodia formation and re-entry across the DREZ. By 3 dpf, the dorsal re-organization of Netrin-expressing glia activates the DCC signaling axis (black), which suppresses stable invadopodia formation at the DREZ, thereby closing the regenerative period. After closure (>3 dpf), the regenerative period can be re-opened through antagonistic manipulations at each step of the DCC signaling axis (green flathead arrows), restoring stable invadopodia formation and central axon penetrance across the DREZ.

By correlating glial cell re-organization with DRG central axon regenerative failure, we provide a cellular mechanism that mediates regenerative period closure. Prior work in mammals has focused on neuronal maturation, largely crediting the resulting intrinsic changes [25,26,32] and diffusely expressed inhibitors such as Nogo, MAG, and CSPGs at the DREZ with the prevention of central axon regeneration [33,78,79]. However, our data in zebrafish suggests that changes in cellular architecture and ligand re-localization between day 2 and 3 of development are critical regulators of the regenerative period. Recent mounting evidence from regenerative studies in mice and zebrafish indicate that glial cells surrounding the DRG undergo distinct changes in response to central axon injury, not only in their organization but in their own expression profiles that negatively impact DRG central axon regeneration [39,40,54]. We propose that peripheral glial re-organization is a key driver of regenerative period closure. During development, we observed an innate dorsal shift of glial cell nuclei surrounding the DRG, leading to a spatial enrichment of *netrin1b* directly along the trajectory of the central axon in 3 dpf animals that persists after axotomy and continues during regenerative attempts. Such glial re-organization appears to reconfigure the guidance landscape and provides a spatial mechanism for regenerative period closure that is not necessarily dependent on expression changes alone. Manipulation of DCC signaling allowed us to re-open the regenerative period after 3 dpf, indicating that DCC signaling is a critical modulator of growth cone cytoskeletal dynamics that closes

the regenerative period of DRG central axons (Fig 9). We demonstrate that by re-opening the regenerative period, circuit and behavioral responses can return to uninjured levels. Collectively, these data reveal a basic and targetable genetic mechanism that closes regenerative periods.

A key component of this research emphasizes the importance of growth cone dynamics and specialized cytoskeletal structures. Unlike filopodia, which are the finger-like exploratory surveyors for growth cone navigation, invadopodia function to degrade extracellular matrices and enable invasion across barriers such as the spinal cord boundary (DREZ) [43,45,48,80,81]. Importantly, previous reports have demonstrated that DCC-cAMP-Rac1 signaling spatiotemporally reduces the formation of stable invadopodia during DRG central axon development until the growth cone reaches the DREZ and invasion is required [44]. Mechanistically, our data presents this DCC signaling pathway as a negative modulator of invadopodia that becomes overactive and leads to regenerative period closure. Based on our data, we propose that glial re-organization and increased expression of *netrin* in the path of the regenerating axon likely creates a dysregulated Netrin-mediated DCC signaling axis when the regenerative period is closed, distinct from early development when such glial re-organization has not yet taken place. This suggests that glial re-organization and increased Netrin presentation create a signaling environment that favors filopodia-mediated exploration but hinders the invasive capacity required for successful regeneration across the DREZ. Long-standing molecular evidence of Netrin modulating filopodia and invadopodia support such a hypothesis [47,82–86]. Furthermore, Rac1 alone has been shown to drive invadopodia disassembly but also regulate filopodia [65,85,87,88]. Netrin-mediated DCC signaling is also known to promote filopodia via the phosphorylation of Ena/VASP by cAMP-dependent PKA activity [80,83,89]. Nevertheless, both filopodia and stable formations of invadopodia are necessary for proper DRG central axon regeneration and re-invasion at the DREZ. Our observations that the antagonistic manipulations of the DCC-cAMP-Rac1 signaling process does not extinguish filopodia or create a global over-stabilizing effect of invadopodia and instead promotes invadopodia stability specifically at the DREZ, indicates that these manipulations are restoring the balance between filopodia and invadopodia present during the development of pioneer central axons. The observation that heterozygous *dcc* mutation can re-open the regenerative period underscores the sensitivity of this process to precise levels of DCC signaling. Reducing DCC levels alone appears sufficient to restore the balance between filopodia and invadopodia, allowing axons to regain their regenerative potential. Taken together, our data provide a genetic mechanism underlying regenerative period closure for centrally-projecting DRG axons.

Cellular architecture and the positioning of cells within tissues shapes the spatiotemporal distribution of membranous and secretory proteins. It has long been established that during development of the CNS, radial glia and astrocytes manipulate the location of molecules such as Slits, Netrins, and Semaphorins to influence axon pathfinding in the brain and spinal cord [90–95]. Further, glia undergo dynamic positional shifts during critical periods of development that correlate with important milestones in synaptic targeting and the establishment of neural networks [17,18,96–99]. After injury, reactive astrocytes have been shown to re-organize, form glial scars, and secrete repulsive molecules to both physically and chemotactically block or redirect regenerating axons [36,100,101]. Our data add additional evidence that mirror these phenomena whereby glial re-organization and receptor/ligand signaling inhibits regeneration. However, our work highlights the importance of these events as an innate developmental process to close the regenerative period rather than a response to injury.

While zebrafish are known for their robust regenerative abilities in other contexts, the swift decline in central axon regeneration capacity mirrors regenerative period closure in other vertebrate systems during development. A similar rapid closure of regenerative periods is observed in corticospinal axons in mice, which can regenerate after spinal cord injury *in utero* until postnatal day 4 (P4) but fail to regenerate by P7, coinciding with a PTEN-dependent downregulation of mTOR which suppresses regenerative potential [23]. Mouse models have demonstrated that regenerative periods in the heart also close by P7, when rodent cardiomyocyte proliferation arrests and key transcriptional and cell cycle regulators are downregulated [21,102]. More recently, other mouse heart studies indicate that closure of the regenerative window begins

between P1 and P2, driven by key changes in the composition and stiffness of the extracellular matrix (ECM) [102,103]. Increased ECM deposition and stiffening seen as early as P2, could be inhibited to rescue cardiac regeneration capacities in P3 mice [103]. These collective findings point to an emerging theme in regenerative periods, where closure is associated with developmentally driven mechanisms to actively suppress regenerative capacity, and that manipulating these mechanisms can re-open these periods.

The idea that regenerative periods are actively suppressed finds parallels in the well-studied phenomenon of critical period closure in neural development. Critical periods are temporal windows of development when neuronal activity can shape cellular morphology and circuit structure. Similar to the glial re-organization that we report closes the regenerative period of DRG axons, circuit plasticity within the CNS is also closed by the re-organization of glia. For example, astrocyte processes dynamically rearrange during critical period closures [16–18]. Molecularly, the astrocytes are reported to terminate the critical period through neuroligin receptors/ligand interactions [18]. While the molecular machinery that closes critical periods currently differ from that of regenerative periods, both phenomena highlight the importance of precisely timed developmental events that constrain plasticity to establish mature neural circuits. Our active suppression hypothesis is consistent with evidence that critical periods can also re-open or remain open, a phenomenon that is often associated with neurocognitive disorders [19,20].

This study highlights a dynamic interplay between glial cell re-organization, spatial distribution of guidance cues, and regulation of growth cone cytoskeletal dynamics to regulate a regenerative period during development. Our findings support the idea that regenerative capacity is not lost, but actively suppressed, offering promising molecular targets to re-open regenerative periods and promote functional recovery after injury.

## Limitations

Our quantifications of glial re-organization were completed by tracking the migration of glial cell nuclei which is indicative but not resolute in establishing the scale of migration for glia. While our transgenic labeling of glial nuclei is also unable to distinguish specific glial cell types, based on our data and the location in the tissue, it is likely that both satellite glia and Schwann cells are the relevant sources of *netrin* that contribute to regenerative period closure. Observations of such peripheral glial populations expressing *netrin* have been made [104,105]. While we do not have the tools to manipulate *netrin* in a cell-specific manner and thus cannot fully test the source of Netrin, ablation of glial cells (nuclear ablation) surrounding the DRG significantly reduced the amount of *netrin1b* transcripts we observed at 3 dpf. These ablations to surrounding glia alone also significantly enhanced DRG central axon regeneration in 3 dpf animals. While outside the scope of this paper, expressing *netrin* ectopically would definitively test the hypothesis that it closes the regenerative period. The analysis in cell-autonomous manipulations was limited to scoring if axons maintained their position at the DREZ, which correlates with regeneration, but does not confirm CNS re-entry directly. Cell-specific experiments also did not distinguish between neuronal and glial molecular function. We were also unable to interrogate the cell-autonomy of cAMP because the genetic modulator is not known. While we demonstrated that the stabilizing effects on invadopodia by rp-cAMP treatment were capable of overcoming cell-specific DCC-induced destabilization and could be reversed by cell-specific activity of Rac1, further characterizations of cAMP in DRG would be beneficial in understanding its role in regenerative period closure.

## Methods

### Ethics Statement

Experimental procedures complied with the NIH guide for care and use of laboratory animals. The University of Notre Dame Institutional Animal Care and Use Committee (IACUC) approved all experiments (protocol 19-08-5464), which is guided by the United States Department of Agriculture, the Animal Welfare Act (USA) and the Assessment and Accreditation of Laboratory Animal Care International (AALAC).

## Experimental model and subject details

**Animal specimens.** Zebrafish used in this study were: *Tg(ngn1:GFP)* [49], *Tg(gfap:NTR-mCherry)* [50], *Tg(sox10:nls-Eos)* [56], *Tg(sox10:Gal4+myl7:GFP)* [59], *Tg(UAS:LifeAct-GFP)* [66], *Tg(NeuroD:Gal4+myl7:GFP)* [46], *Tg(UAS:GCaMP6s)* [75], and *Tg(sox10:mRFP)* 76. Stable mutant lines were generated by crossing *dcc^zm130198^* [60] into relevant transgenic backgrounds. All embryos were generated via pairwise matings and grown at the recommended 29°C in our satellite fish facility incubator. After 24 hpf, zebrafish embryos were maintained in PTU (0.0003%) to reduce pigmentation for *in vivo* imaging. The age of our animals for experiments were determined by developmental stages and hpf (hours post fertilization) [106] (Table 1).

***In Vivo* Imaging and Timelapse Imaging.** Animals were anesthetized using veterinary grade 3-aminobenzoic acid ester (Tricaine, Syndel) for mounting purposes. Animals were mounted laterally (their right side) in glass bottomed dishes and covered in 0.8% low melt agarose [52,107]. For pre-injury and 24-hpi imaging, 1x Tricaine in egg water was added to the dish. Pre-, post-, and 24 hr post-injury images consisted of a 40 μm z stack (1 μm step size). For overnight imaging experiments, 1x Tricaine in egg water was added to the dish. All of the images in this study were obtained with our spinning disk confocal microscopes custom built by 3i technology (Denver, Colorado) which contains: Zeiss Axio Observer Z1 Advanced Mariana Microscope, X-cited 120LED White Light LED System, filter cubes for GFP and mRFP, a motorized X, Y stage, piezo Z stage, 20x Air (0.50NA), 63x (1.15NA), 40x (1.1NA) objectives, CSU-W1 T2 Spinning Disk Confocal Head (50 μm) with 1x camera adapter, and Prime 95B back illuminated CMOS camera by Teledyne, dichroic mirrors for

**Table 1. Key resources table.**

| Reagent or Resource | Source | Identifier |
|---|---|---|
| **Chemicals** | | |
| Rp-cAMP | Fisher Scientific | #1168145UMOL |
| Sp-cAMP | Fisher Scientific | #11681510UMO |
| NSC23766 | Milipore Sigma | #553502 |
| **HCR** | | |
| HCR Probes and Reagents | Molecular Instruments | HCR Gold RNA-FISH Kit |
| **Immunohistochemistry** | | |
| Anti-Cortactin (mouse) | Sigma | SAB4500766 |
| Anti-Sox10 (rabbit) | Binari et al (2013)[64] | ZDB-ATB-130417–3 |
| Alexa Fluor 647 anti-mouse | Invitrogen | A-21235 |
| Alexa Fluor 647 anti-rabbit | Invitrogen | A-21244 |
| **Transgenesis** | | |
| *psCJS16 (UAS:dcc-TdTomato)* | Kikel-Coury et al (2021)[44] | *psCJS16* |
| *UAS:pa-Rac1-mCherry* | Wu et al (2009)[69] | Addgene #41878 |
| **Mutagenesis** | | |
| *dcc^zm130198^* | Jain et al (2014)[60] | ZDB-ALT-150211–1 |
| **Zebrafish Lines** | | |
| *Tg(ngn1:GFP)* | McGraw et al (2008)[49] | ZDB-FISH-150901–27602 |
| *Tg(gfap:NTR-mCherry)* | Smith et al (2016)[50] | ZDB-TGCONSTRCT-160630–2 |
| *Tg(sox10:nls-Eos* | McGraw et al (2012)[56] | ZDB-TGCONSTRCT-110721–2 |
| *Tg(sox10:Gal4+myl7:GFP)* | Hines et al (2015)[59] | ZDB-TGCONSTRCT-140722–3 |
| *Tg(UAS:LifeAct-GFP)* | Helker et al (2013)[66] | ZDB-TGCONSTRCT-130624–2 |
| *Tg(NeuroD:Gal4+myl7:GFP)* | Nichols et al (2019)[46] | ZDB-TGCONSTRCT-191209–8 |
| *Tg(UAS:GCaMP6s)* | Thiele et al (2014)[75] | ZDB-TGCONSTRCT-140811–3 |
| *Tg(sox10:mRFP)* | Kucenas et al (2008)[76] | ZDB-TGCONSTRCT-080321–2 |

445, 515, 561, 405, 488, 561, 640 excitation, laser stack with 405 nm, 445 nm, 488 nm, 561 nm, and 637 nm laser stack FiberSwitcher, photomanipulation from vector high speed point scanner ablations at diffraction limited capacity, Ablate! Photoablation System (532 nm pulsed laser, pulse energy 60J @ 200 Hz). Timelapse images were collected every 5 min for 22–24 hrs starting after central axon axotomies using Slidebook software. Images were processed using ImageJ and Adobe Illustrator to enhance brightness and contrast.

**Axotomy and glia ablations.** The Ablate! photoablation system provided by 3i was used for all axotomies and glial ablations in this study [42,108]. This lesioning system targets select z-planes delivered to curser-selected X, Y positions. The laser is diffraction-limited with an adjustable roster block size, controlling the size of the injury site [108]. This, coupled with the scalable laser power, mitigates injury to surrounding tissues. To axotomize the centrally-projecting DRG axons, the axon's widest section at the midway point between the DREZ and DRG was put in focus and then the z-plane was backed out -2 µm [42,108]. The laser power and number of laser pulses were consistent across experiments in this study. Full transection of the nerve was confirmed by long-term loss of fluorescence at the site of injury, the appearance of slack along the proximal end of the central projection, as well as the formation of a retraction bulb at the distal end of the severed projection [42]. The same lesioning system, settings, and routine were used to ablate the photoconverted glial cell nuclei surrounding the DRG. Long-term loss of fluorescence and appearance of debris were used to confirm cell death.

**Orthogonal Displacement Quantifications.** In Slidebook software, 3view settings enabled orthogonal sideview images (rotated 90°) for select X, Y locations. These images were converted to a 24-bit RGB-Tiff file and exported to ImageJ to obtain fluorescent profiles of the distal tip of the regenerative axon (*ngn1*:GFP+) and the glial limitans of the spinal cord boundary (*gfap*-mCherry+) along an 8 µm line drawn across their junction (Fig 1) [43]. If the peak of the axon's fluorescence occurred medial, or inside, the spinal cord boundary fluorescent peak the axon was characterized as entered. If the axon's fluorescent peak was lateral (outside) to the spinal cord boundary fluorescent peak, then it was characterized as not entered.

**Glia Re-Organization and Glia Ablation.** Glia re-organization was evaluated in *Tg(ngn1:GFP)* and *Tg(sox10:nls-Eos)* animals where we could visualize *ngn1*+ DRG and photoconverted *sox10*+ glial nuclei surrounding the DRG. Photoconversions were completed by exposing the entire dish of animals to LED-UV Array light 3 times (5 sec each) prior to mounting using LOCTITE LED Flood System (97070/97071). Central axon axotomy of a single DRG (#4–6) per animal was performed and then was timelapse imaged for 22–24 hours (40 µm z stack, 5 min intervals) to capture both the DRG and photoconverted glial nuclei. Maximum z projections were made for each timelapse in Slidebook software and exported to ImageJ as 16-bit Tiff files. Analysis of glia re-organization was done by using ImageJ's MTrackJ feature, where we obtained the X, Y position of both the growth cone and the dorsal most glial nucleus relative to the X, Y position of dorsal border of the DRG. Migration distances were measured when regenerative neurite extension began and lasted for 140 timepoints (5 min intervals, ~11.6 hours). The growth cone and glia nucleus Y positions were subtracted from the Y position of the dorsal border of top DRG soma in each of the 140 timepoints analyzed. These 140 Y position measurements for the growth cone and glia nucleus were then compiled by age group. In this analysis, only the glia that were directly surrounding the DRG (satellite glia) were measured. For glia ablation experiments, prior to central axon axotomy, all neighboring glial cells surrounding the DRG and central axon were ablated with the Ablate lesioning system in 3 dpf *Tg(ngn1:GFP)*;*Tg(sox10:nls-Eos)* animals [52,108]. Glial nuclei death was confirmed by long-term loss of fluorescence and/or presence of debris. These animals and their DRG central axons were re-imaged at 24 hpi to evaluate sustained regrowth back to the DREZ. For these experiments, animals that underwent neuronal cell death were not included in regeneration analyses.

**Whole-Mount Larval Zebrafish Hybridized Chain Reaction (HCR).** HCR RNA in whole mount fixed zebrafish embryos was done following Molecular Instruments Inc. protocol and their reagents and probes. In brief, *Tg(ngn1:GFP)* animals were fixed with 4% paraformaldehyde in PBS for 1 hour at 48 and 72 hpf, as well as 52 hpf (4 hpi), 76 hpf (4 hpi), and 82 hpf (10 hpi). Animals were dehydrated with 100% Methanol (MeOH) and stored at -20°C. Animals were rehydrated

with a series of washes with MeOH and PBS with 0.1% Tween 20 (PBST); 75% MeOH and 25% PBST, 50% MeOH and 50% PBST, 25% MeOH and 75% PBST, and then 100% PBST [72,109]. Animals were refixed in 4% PFA and washed with PBS for 20 min. Hybridization was done with either a *dcc* or *netrin1b* probe (1:250, 2 µL, B1). Amplification was performed with B1-h1-647 and B1-h2-647 (1:50, 10 µL) [72,109]. HCR fish were protected from light while being washed with and then stored in 5x sodium chloride sodium citrate (SSC) with 0.1% Tween 20 (4°C) until imaging. Images consisted of DRG 3–6 with an 80 µm z stack. Injured animals had previously undergone central axon axotomy to DRG #4 in each animal. In glia ablated animals, each animal underwent ablation of photoconverted *sox10*⁺ nuclei neighboring and dorsal to DRG #5 prior to fixing 4 hours post ablation.

The number of *dcc* puncta (647 channel) found within the confines of the DRG neurons (488 channel) was divided by the total area of the DRG (traced in Slidebook software while scrolling through the z stack). To analyze *netrin1b* HCR, the border of the *netrin1b* puncta aggregations was traced from the dorsal edge of the DRG soma to the DREZ (characterized by the dorsal longitudinal fasciculus (DLF)). The mean grey value was calculated by dividing the fluorescence intensity of 647 channel by the area of the tracing. Adjacent to each DRG, where no *netrin1b* puncta were, a background mean grey value was obtained. The adjacent background was subtracted from the *netrin1b* mean grey value of each corresponding DRG analyzed. Data from HCR animals were compiled according to their age and hpi groupings. Comparisons between groups were made with Kruskal-Wallis One Way ANOVA (p = 0.05) for both *dcc* and *netrin1b* HCR. In the injured glial ablation context, DRG neuronal death excluded the animal from inclusion in the analysis. The mean grey value for *netrin1b* expressed in the spinal floor place was calculated from the flanking regions of each DRG using a square of 200 µm².

**Drug Treatments.** The pharmacological manipulations of the DCC-cAMP-Rac1 signaling process were: sp-cAMP [63] (Fisher Scientific, catalog #11681510UMO), rp-cAMP [61] (Fisher Scientific, catalog #1168145UMOL), NSC23766 [62] (Milipore Sigma, catalog #553502). Stock solutions of these reagents were stored at -20°C with concentrations of 1 mM (sp-cAMP), 60 µM (rp-cAMP), and 25 µM (NSC23766). For all treatments, animals were bathed in drug solutions of 100 µM (sp-cAMP), 25 µM (rp-cAMP), and/or 1 µM (NSC23766) for ~10–20 min before undergoing central axon axotomy and then kept in those treatments for 24 hours [44]. All drug treatments were mixed into a DMSO and PTU solution to achieve the desired working concentration (listed above) in 1% DMSO. 1X Tricaine was included in these mixtures only for axotomies and/or long-term timelapse imaging.

**Actin Dynamics and Quantifications.** For all actin quantifications, *Tg(sox10:Gal4 + myl7:GFP);(UAS:LifeAct-GFP)* animals underwent central axon axotomy of a single DRG axon bundle at 3 dpf followed by a 22–24 hr timelapse (5 min intervals, 40 µm z stacks with a step size of 1 µm step size). Maximum projections were made to collapse the z-planes and quantifications were made in ImageJ. During quantification, only DRG where the central axon growth cone was visible during the entire timelapse were scored. Using the MTrackJ feature of ImageJ, in each frame after axon extension began, the center of the growth cone was selected [42–44]. At the end of the timelapse the integrated density of the growth cones fluorescence from each frame was exported into Microsoft Excel. For each timepoint analyzed, a background sample of the integrated density was taken dorsal to the spinal cord to subtract from the corresponding timepoint of the growth cone fluorescent measurements [42–44]. The average growth cone fluorescent intensity was calculated from all the frames analyzed in the timelapse to create a threshold line to determine invadopodia state. Any timepoint where the growth cone's actin fluorescence was above the average was labeled, and an event of 4 or more consecutive timepoints above threshold (equating to 20 min or more) was characterized as an invadopodia formation. The durations of all these invadopodia formations were compiled according to the genetic and/or treatment group. Separately, this same compilation was performed for only the invadopodia formations (actin peaks >3 consecutive timepoints) that occurred when the growth cone was at the DREZ, characterized by the dorsal longitudinal fasciculus (DLF). Comparisons between groups in both compilations were made via Brown-Forsythe and Welch ANOVA (p = 0.05).

**Invadopodia vs filopodia characterizations.** In timelapse movies of *Tg(sox10:Gal4 + myl7:GFP);(UAS:LifeAct-GFP)* animals, characterizations of growth cone morphologies were made with the following parameters. Timepoints

displaying robust, long-duration actin accumulations at the center of the regenerative cone with basal protrusions toward the DREZ were characterized as stable invadopodia formations [44]. Filopodia-based growth cones were characterized by low LifeAct intensity measurements accompanied by long, finger-like projections [44]. To quantify the morphological differences between these two structures of the growth cone, the orthogonal widths of filopodia and invadopodia were obtained from 25 growth cone measurements in 4 and 5 biological replicates, respectively. These measurements consisted of drawing a line across the orthogonal view of the growth cone to obtain the width (in µm) of the LifeAct fluorescence. These widths were compared using Mann-Whitney T-test (p = 0.05).

**Cell-specific DCC expression.** The expression plasmid for *UAS*-driven DCC-tdTomato was obtained from a previously published study in zebrafish DRG [44]. The plasmid was injected at the one cell stage. At 72 hpf, mounted animals were screened on our spinning disk confocal microscope for tdTomato⁺ puncta inside DRG. One DCC-tdTomato⁺ DRG per animal underwent central axon axotomy followed by a 22–24 hpi timelapse. All analyses of actin dynamics (described above) were done prior to animals being genotyped for *dcc^zm130198* [60]. For inclusion, the axotomized DRG had to undergo a significant re-extension (>15 hrs) and the growth cone must have been visible the entire timelapse. Comparisons were made with Brown-Forsythe and Welch ANOVA test (p = 0.05).

**Photoactivatable Rac1 *(UAS:pa-Rac1-mCherry)*.** The expression plasmid for *uas:pa-Rac1-mCherry* was obtained from a previously published study in zebrafish DRG [43,44,69]. The plasmid was injected at the one-cell stage. At 3 dpf, mounted animals were screened for pa-Rac1-mCherry⁺ DRG on our spinning disk confocal microscope, wherein one mCherry⁺ DRG underwent central axon axotomy per animal. The pa-Rac1 is activated by 445 nm light, therefore, both uninjected animals exposed to 445 nm light and pa-Rac1-mCherry⁺ DRG unexposed to 445 nm light were used as controls. After axotomy, a 22–24 timelapse was taken. Analysis of actin dynamics followed the same procedure as described previously. Regenerating central axons had to undergo a re-extension of at least 15 hrs with the growth cone visible the entire timelapse to be included for analysis. Comparisons were made with Brown-Forsythe and Welch ANOVA test (p = 0.05).

**Genotyping *dcc^zm130198*.** After performing experimental imaging, the larvae were lysed using a mixture of 20 mM Tris buffer (pH = 8), 4 mM Ethyenediamine Tetraacetate Acid in 0.4% Triton-X100 nuclease free water and proteinase K at 50°C for 12 hours and then 95°C for 15 minutes. Primers provided by Integrated DNA Technologies were used at 10 mM concentrations with the sequences 5'-GCGCAGCTGTCTGTCAGTAG-3' (Forward), 5'-GACGCAGGCGCATAAAATCAGTC-3' (Reverse), and 5'-CGCAGATCTGTGCGTAGGAGAGC-3' (ZM130198 Forward) [60]. Polymerase Chain Reaction samples were run in 1% agarose gel with a 1kb+ ladder, and WT genotype corresponded to bands of 203 bp while mutant genotype corresponded to bands of 766 bp [60].

**Immunohistochemistry.** The primary antibody for Anti-Cortactin IHC is from Sigma (1:50, catalog number: 05–180-I-100UL). The secondary antibody used was Alexa Fluor 647 goat anti-mouse (1:600, Invitrogen, catalog number: A-21235). The primary antibody for Sox10 was rabbit (1:5000, ZFIN ID: ZDB-ATB-130417-3) [64]. The secondary antibody was Alexa Fluor 647 goat anti-rabbit (1:600, Invitrogen A-21244). Animals were fixed using 4% paraformaldehyde in PBSTx (phosphate buffered saline (PBS) with 1% Triton-X100) at 4°C for 24 hours. The fixed larvae were washed in PBSTx, deionized water with 1% Triton-X100, and acetone for 5 min each [110]. The larvae were incubated in cold acetone at -20°C for 10 min. Larvae were washed 3 times with PBSTx for 5 min and incubated in with 5% goat serum in PBSTx for 1 hour at room temperature [110]. The larvae were then incubated with the Anti-Cortactin or Sox10 primary antibody solution (PBSTx and 5% goat serum) at 4°C overnight. After washing 3 times with PBSTx for 30 min each and a wash with PBSTx for 1 hour, the larvae were incubated with the secondary antibody solution at 4°C overnight [110]. After 3 washes with PBSTx for 1 hour each, larvae were stored in 50% glycerol in PBS at 4°C until imaging. These larvae and their DRG were imaged using the same protocol as previously described for *in vivo* imaging.

**Calcium imaging and analysis.** For the calcium imaging experiment to assess functional circuits between the DRG and spinal cord, 5 dpf *Tg(NeuroD:Gal4 + myl7:GFP);(UAS:GCaMP6s)* and *Tg(sox10:mRFP)* animals were individually

mounted in glass bottom dishes in a thin layer of 0.8% low melting point agarose [71,72]. The layer of agarose had to be thin enough that a curve around the body of the animal could be detected by light diffraction to ensure rapid sensation of cold water. Animals were anesthetized only for mounting and afterward were given 20 mins to recover to ensure responsiveness to cold exposure. On a spinning dish confocal microscope, a GCaMP6s⁺ DRG was located between DRG #6–9 and positioned 4 µm from the top of a 40 µm z-stack (2 µm step size). This z-stack size ensured that the DRG and roughly half of the spinal cord could be included while also maintaining a rapid turnover between captures. The water was aspirated out of the imaging dish and a rapid, 24 frame (120 sec, 5 sec intervals) timelapse was started to obtain baseline GCaMP6s fluorescence. After 39 seconds (just before the 8th frame) 23°C water was added to the imaging dish, where it remained for 5 frames (25 sec) before being aspirated out. 39 seconds after the 23°C addition (just before the 16th frame) 4°C water was added to the imaging dish, where it remained for the rest of the timelapse. This process was done carefully to not disturb the dish and the positioning of the DRG and spinal neurons in the z-stack. For injured animals, *Tg(NeuroD:Gal4 + myl7:GFP);(UAS:GCaMP6s)* and *Tg(sox10:mRFP)* animals were used to perform 8 series axotomies to DRG #4–11 at 3 dpf. These animals were then incubated in DMSO as a vehicle control, PTU only as a no treatment group, or pharmacological manipulations cAMP or Rac1 for 24 hpi, followed by another 24 hpi in PTU only to further recover. Heterozygous *dcc* mutants were also only incubated in DMSO and were obtained by mating *Tg(NeuroD:Gal4 + myl7:GFP);(UAS:GCaMP6s)* and *Tg(sox10:mRFP);dcc⁺/⁻* animals, however genotyping was only completed after analyses were completed. At 48 hpi (5 dpf) the animals underwent the calcium imaging assay described above.

To analyze GCaMP6s transients, maximum z-projections were made for each animal and exported as a 16-bit TIFF to be opened in ImageJ. GCaMP6s⁺ cell somas visible the entire timelapse were used as landmarks to align the frames of the timelapse to correct for any movements of the animal during imaging [72]. Then, using the ROI manager plugin of ImageJ, the DRG neurons and 30 spinal neuron cell somas were traced in the 16th frame (the first capture immediately following cold-water immersion). To limit bias, the 30 spinal neurons were analyzed regardless of their apparent brightness in the 16th frame. The integrated density of each DRG neuron and spinal neuron tracing was exported to a spreadsheet, where the average GCaMP6s fluorescence and standard deviation was calculated in order to calculate a z-score for each cell in each frame of the timelapse. A z-score of 2.0 or more was characterized as a neuronal activation event [72]. For inclusion, we confirmed that at least one DRG neuron per ganglia rapidly activated in the 16th frame (first cold-water capture). After this confirmation, the proportion of spinal neurons that activated in the first cold-water capture (synchronous activity with the DRG) was compared to the proportion of spinal neurons that only had a z-score >2.0 in the 17th frame or later (delayed activity) and the proportion of spinal neurons that never had a z-score of 2.0 or more (no active event). The percentage of spinal neuron responses were compiled by treatment or genotype group. The spinal neuron responses between groups were compared on the basis of rapid (synchronous activity with the DRG) and delayed activity using Fisher's Exact test (p = 0.05). This comparison was also done for the DRG neurons per animal across groups.

**Behavioral Assay.** In a petri dish on a Zeiss Axio Observer microscope, a pipet was used to add a sample of 23°C water to create a small testing arena that ensured that the animal could not swim out of view. Using the brightfield settings, a 600-frame movie was started (30 sec, 50 ms intervals) and a 5dpf *Tg(ngn1:GFP);(gfap:NTR-mCherry)* animal was gently added to the testing arena. Using a separate dish to prevent temperature fluctuations, a 4°C testing arena was made, and a new movie was started. The animal was then added to this 4°C water to evaluate its shivering response [42,44]. In analysis of the behavioral assay, movies were opened in ImageJ where the first 400 frames (20 sec) after animal was fully immersed in the cold water was analyzed and the number of frames the animal spent shivering, characterized by head and tail movements without forward locomotion, was quantified. The number of frames spent shivering over the 400 frames analyzed were represented as a percentage per animal and compiled in groups according to injured status, treatment, and/or genotype. In the injured context, 3 dpf *Tg(ngn1:GFP);(gfap:NTR-mCherry)* animals underwent series axotomies to DRG #4–11, followed by 24 hpi incubation in DMSO as vehicle control or pharmacological

manipulations of cAMP or Rac1 in WT and *dcc*[+/−] animals. Combination treatments for epistatic analyses were also performed. After the 24 hpi of treatment, animals were allowed to recover for an additional 24 hours. At 48 hpi, the animals were re-imaged to assess how many DRG central axons regenerated via orthogonal displacement quantifications. These animals were then unmounted, given 30 mins to recover from the anesthetic and then used in the behavioral assay described above. Uninjured animals were also given the same pharmacological manipulations in the same time frame as injured animals. To limit bias, the behavioral assay was analyzed prior to orthogonal displacement quantifications. All genotyping was performed after both analyses were completed. The percentage of time spent shivering and central axon re-entry rates were compared between groups using Kruskal-Wallis One-Way ANOVA (p = 0.05).

## Statistical analysis

Statistical analyses were completed with Prism. No statistical methods were used to predetermine sample sizes, but sample sizes were similar to previous publications. Statistical tests were performed with biological replicates (animals) rather than technical replicates. No data points were excluded for these analyses. Characteristics for inclusion for analyses were predetermined for calcium imaging, actin dynamics, and glia ablations (DRG central axon regeneration) as discussed in those sections. Prior to all experiments, healthy animals were selected at random. Each experiment in each condition was repeated at least once to confirm results. All data collected and analyzed are presented in this study.

## Software

Slidebook, Prim, ImageJ, Microsoft Excel, and Adobe Illustrator were used to acquire, analyze and compile figures. Microsoft Word was used in the construction of this manuscript. Google Gemini was used to edit the manuscript. Mendeley was used to create and organize citations.

## Supporting information

**S1 Data. Excel sheet with tabs of raw data information for each graph, tabulation, and statistical test of this manuscript.**
(XLSX)

**S1 Fig. A, C and E)** Max projection images of *Tg(ngn1:GFP)* DRG and central axons pre-axotomy (0 hpi), post-axotomy (0.1-0.25 hpi) and 23–24 hpi. Scale bar = 10 μm. Yellow arrows point to bifurcated axons. The treatment conditions are listed above images. **B, D and F)** DRG axons' orthogonal images and orthogonal displacement quantifications. White arrowheads indicate the axons' location in the orthogonal images. **G)** Representative images of Sox10 staining (IHC) labelling glial nuclei (pink) in 3 dpf *Tg(ngn1:GFP)* animals. These images represent animals that were fixed: Uninjured at 72 hpf (WT), 12 hpi in *dcc*[+/−] animals treated with 1% DMSO, 12 hpi treated with rp-cAMP (WT), and 12 hpi treated with NSC23766 (WT). Scale bars = 10 μm. Yellow arrowheads point to the dorsal-most glial nuclei. **H)** Glial Organization target graphs of the distance between the dorsal border of the DRG (black X) and the center of the dorsal-most glial nuclei across each biological replicate (pink circles) in each group. **I)** DRG to Glial Nucleus Position Difference graph display the distances between the DRG and dorsal-most glial nuclei (± SEM) per group. The average distances ± SEM are: Uninjured (72 hpf) 10.86 ± 2.404 μm (n = 8), 12 hpi WT (DMSO) 9.472 ± 1.614 μm (n = 10), 12 hpi *dcc*[+/−] (DMSO) 11.59 ± 1.912 μm (n = 7), 12 hpi rp-cAMP 10.26 ± 2.256 μm (n = 7), 12 hpi NSC23766 10.77 ± 1.803 μm (n = 7). Kruskal-Wallis One-Way ANOVA (p = 0.05). Raw data information for this figure can be found in S1 Data.
(TIF)

**S2 Fig. A)** Max projections of *Tg(sox10:Gal4 + myl7:GFP);(UAS:LifeAct-GFP)* DRG post-axotomy (0.1 hpi) and growth cones during 24 hpi timelapses. White dashed boxes indicate the growth cone positioned at the DREZ, yellow arrows point to bifurcated axons. Treatment conditions for each are listed at the top of each panel. Quantifications of growth cone

(GC) LifeAct intensity measurements in 3 regenerative growth cones under antagonistic treatment (rp-cAMP + NSC23766) and agonistic treatments (sp-cAMP in WT and $dcc^{+/-}$ animals) over the 22–24 hpi timelapses. The grey portions of these line graphs indicate when the growth cone is navigating, and the black portions indicate when the growth cone was at the DREZ. Colored brackets under peaks of LifeAct fluorescence represent their duration. See key above. Raw data information for this figure can be found in S1 Data.
(TIF)

**S3 Fig. A)** Quantification of LifeAct peak durations (20 min or more) occurring in the growth cone throughout the entire 22–24 hour timelapse in DCC-tdTomato expressing DRG and uninjected controls. Uninjected $dcc^{+/-}$ n = 3, Uninjected wt n = 4, DCC-tdTomato$^+$ $dcc^{+/-}$ n = 4, DCC-tdTomato$^+$ wt siblings n = 4, DCC-tdTomato$^+$ $dcc^{+/-}$ animals in rp-cAMP treatment n = 5. **B)** Quantification of LifeAct peak durations (20 min or more) occurring in the growth cone throughout the entire 22–24 hour timelapse in pa-Rac1-mCherry$^+$ DRG and uninjected controls. Uninjected rp-cAMP exposed to 445 nm light n = 6, pa-Rac1 expressing rp-cAMP treated unexposed to 445 nm light n = 6, pa-Rac1 expressing rp-cAMP treated exposed to 445 nm light n = 7. For (A) and (B), comparisons between groups were made with Kruskal-Wallis One-Way ANOVA tests. Raw data information for this figure can be found in S1 Data.
(TIF)

**S4 Fig. Heat map of the p-values obtained from Kruskal-Wallis One-Way ANOVA test of the percentage of time spent shivering between series axotomized treatment groups 48 hpi.** The raw data information for this figure can be found in S1 Data.
(TIF)

**S1 Movie. Timelapse of axon and glial nucleus migration in a 2 dpf *Tg(ngn1:GFP);Tg(sox10:nls-Eos)* animal after DRG central axon axotomy.** A red open circle indicates the axons growth cone position, and the yellow open circle indicates the dorsal-most glial nucleus around the DRG, beginning when axon re-extension begins for 140 timepoints (5 min intervals). Video rate is 10 frames per second. Scale bar = 10 μm.
(MOV)

**S2 Movie. Timelapse of axon and glial nucleus migration in a 3 dpf *Tg(ngn1:GFP);Tg(sox10:nls-Eos)* animal after DRG central axon axotomy.** A red open circle indicates the axons growth cone position, and the yellow open circle indicates the dorsal-most glial nucleus around the DRG, beginning when axon re-extension begins for 140 timepoints (5 min intervals). Video rate is 10 frames per second. Scale bar = 10 μm.
(MOV)

**S3 Movie. Timelapse of DMSO control treated growth cone in a 3 dpf *Tg(sox10:Gal4+ myl7:GFP);(UAS:LifeAct-GFP)* animal.** The timelapse starts 0.1 hours after central axotomy and covers 22 hours. A red open circle denotes the regenerative growth cone once axon re-extension begins. Video rate is 10 frames per second. Scale bar = 10 μm.
(MOV)

**S4 Movie. Timelapse of a growth cone in a 3 dpf *Tg(sox10:Gal4+ myl7:GFP);(UAS:LifeAct-GFP); dcc$^{+/-}$* animal treated with DMSO.** The timelapse starts 0.1 hours after central axon axotomy and covers 22 hours. A red open circle denotes the regenerative growth cone once axon re-extension begins. Video rate is 10 frames per second. Scale bar = 10 μm.
(MOV)

**S5 Movie. Timelapse of growth cone in after rp-cAMP treatment of a 3 dpf *Tg(sox10:Gal4+ myl7:GFP);(UAS:LifeAct-GFP)* animal.** The timelapse starts 0.1 hours after central axotomy and covers 22 hours. A red open circle denotes the regenerative growth cone once axon re-extension begins. Video rate is 10 frames per second. Scale bar = 10 μm.
(MOV)

**S6 Movie. Timelapse of growth cone in after NSC23766 treatment of a 3 dpf *Tg(sox10:Gal4+ myl7:GFP);(UAS:LifeAct-GFP)* animal.** The timelapse starts 0.1 hours after central axotomy and covers 23 hours. A red open circle denotes the regenerative growth cone once axon re-extension begins. Video rate is 10 frames per second. Scale bar = 10 μm.
(MOV)

**S7 Movie. Timelapse of growth cone in after sp-cAMP treatment of a 3 dpf *Tg(sox10:Gal4+ myl7:GFP);(UAS:LifeAct-GFP)* animal.** The timelapse starts 0.1 hours after central axotomy and covers 23 hours. A red open circle denotes the regenerative growth cone once axon re-extension begins. Video rate is 10 frames per second. Scale bar = 10 μm.
(MOV)

**S8 Movie. Timelapse of a growth cone of a DCC-tdTomato[+] DRG in a 3 dpf *Tg(sox10:Gal4+ myl7:GFP);(UAS:LifeAct-GFP); dcc[+/−]* animal treated with only DMSO.** The timelapse starts 1 hour after central axotomy and covers 22 hours. A red open circle denotes the regenerative growth cone once axon re-extension begins. Video rate is 10 frames per second. Scale bar = 10 μm.
(MOV)

**S9 Movie. Timelapse of a growth cone of a DCC-tdTomato[+] DRG in a 3 dpf *Tg(sox10:Gal4+ myl7:GFP);(UAS:LifeAct-GFP)* animal treated with only DMSO.** The timelapse starts 0.1 hours after central axotomy and covers 23 hours. A red open circle denotes the regenerative growth cone once axon re-extension begins. Video rate is 10 frames per second. Scale bar = 10 μm.
(MOV)

**S10 Movie. Timelapse of a growth cone of a DCC-tdTomato[+] DRG in a 3 dpf *Tg(sox10:Gal4+ myl7:GFP);(UAS:LifeAct-GFP);dcc[+/−]* animal treated with rp-cAMP.** The timelapse starts 0.1 hours after central axotomy and covers 23 hours. A red open circle denotes the regenerative growth cone once axon re-extension begins. Video rate is 10 frames per second. Scale bar = 10 μm.
(MOV)

**S11 Movie. Timelapse of a growth cone of a pa-Rac1-mCherry[+] DRG in a 3 dpf *Tg(sox10:Gal4+ myl7:GFP);(UAS:LifeAct-GFP)* animal treated with rp-cAMP that was unexposed to the 445 nm activating light.** The timelapse starts 0.1 hours after central axotomy and covers 23 hours. A red open circle denotes the regenerative growth cone once axon re-extension begins. Video rate is 10 frames per second. Scale bar = 10 μm.
(MOV)

**S12 Movie. Timelapse of a photoactivated-Rac1 growth cone of pa-Rac1-mCherry[+] DRG in 3 dpf *Tg(sox10:Gal4+ myl7:GFP);(UAS:LifeAct-GFP)* animal treated with rp-cAMP that was exposed to the 445 nm activating light (every 5 min).** The timelapse starts 0.1 hours after central axotomy and covers 23 hours. A red open circle denotes the regenerative growth cone once axon re-extension begins. Video rate is 10 frames per second. Scale bar = 10 μm.
(MOV)

**S13 Movie. Timelapse of an uninjured 5 dpf *Tg(NeuroD:Gal4+ myl7:GFP);(UAS:GCaMP6s)* animal that displays the response of DRG and spinal neurons to the evoked calcium imaging assay.** A red open circle denotes the DRG activated (z-score >2.0) in the first capture after the animal is immersed in the cold-water (4°C) stimulus. All spinal neurons activated (z-score >2.0) in the first cold-water capture, synchronized with the DRG. The timelapse consists of 24 captures (5 sec intervals) for 120 seconds. Video rate is 5 frames per second. Scale bar = 10 μm.
(MOV)

**S14 Movie. Timelapse of a series axotomized (48 hpi)** *5 dpf Tg(NeuroD:Gal4⁺ myl7:GFP);(UAS:GCaMP6s)* **animal treated with only DMSO.** The timelapse displays the response of DRG and spinal neurons to the evoked calcium imaging assay. A red open circle denotes the DRG activated (z-score >2.0) in the first capture after the animal is immersed in the cold-water (4°C) stimulus. Other colored open circles denote spinal neurons that activated (z-score >2.0) in a delayed manner, after the first cold-water capture, unsynchronized with the DRG. The timelapse consists of 24 captures (5 sec intervals) for 120 seconds. Video rate is 5 frames per second. Scale bar = 10 µm. (MOV)

**S15 Movie. Timelapse of a series axotomized (48 hpi) 5 dpf** *dcc*+/⁻; *Tg(NeuroD:Gal4+ myl7:GFP);(UAS:GCaMP6s)* **animal treated with only DMSO.** The timelapse displays the response of DRG and spinal neurons to the evoked calcium imaging assay. A red open circle denotes the DRG activated (z-score >2.0) in the first capture after the animal is immersed in the cold-water (4°C) stimulus. All spinal neurons activated (z-score >2.0) in the first cold-water capture, synchronized with the DRG. The timelapse consists of 24 captures (5 sec intervals) for 120 seconds. Video rate is 5 frames per second. Scale bar = 10 µm. (MOV)

**S16 Movie. Timelapse of a series axotomized (48 hpi) 5 dpf** *Tg(NeuroD:Gal4⁺ myl7:GFP);(UAS:GCaMP6s)* **animal treated with rp-cAMP.** The timelapse displays the response of DRG and spinal neurons to the evoked calcium imaging assay. A red open circle denotes the DRG in the first capture after the animal is immersed in the cold-water (4°C) stimulus. All spinal neurons but one activated (z-score >2.0) in the first cold-water capture, synchronized with the DRG. The timelapse consists of 24 captures (5 sec intervals) for 120 seconds. Video rate is 5 frames per second. Scale bar = 10 µm. (MOV)

## Acknowledgments

We acknowledge the valuable discussions with past and present members of the Smith Lab. We also appreciate Deborah Bang and Matthew Lewis for their dedicated care of the zebrafish.

## Author contributions

**Conceptualization:** Cody J. Smith.

**Formal analysis:** Jacob Hammer.

**Funding acquisition:** Cody J. Smith.

**Investigation:** Jacob Hammer.

**Methodology:** Jacob Hammer.

**Project administration:** Cody J. Smith.

**Validation:** Jacob Hammer.

**Visualization:** Jacob Hammer.

**Writing – original draft:** Jacob Hammer, Cody J. Smith.

**Writing – review & editing:** Jacob Hammer, Cody J. Smith.

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
