## [Decision Letter · Decision Letter 0]

5 Oct 2025

PGENETICS-D-25-00917

The regenerative period of somatosensory nerves is closed by a DCC signaling axis

PLOS Genetics

Dear Dr. Smith,

Thank you for submitting your manuscript to PLOS Genetics. After careful consideration, we feel that it has merit but does not fully meet PLOS Genetics's publication criteria as it currently stands. Therefore, we invite you to submit a revised version of the manuscript that addresses the points raised during the review process.

Please submit your revised manuscript within 30 days Nov 04 2025 11:59PM. If you will need more time than this to complete your revisions, please reply to this message or contact the journal office at plosgenetics@plos.org. Please include the following items when submitting your revised manuscript:

We look forward to receiving your revised manuscript.

Kind regards,

Benjamin Podbilewicz

Academic Editor

PLOS Genetics

Fengwei Yu

Section Editor

PLOS Genetics

Aimée Dudley

Editor-in-Chief

PLOS Genetics

Anne Goriely

Editor-in-Chief

PLOS Genetics

**Additional Editor Comments:**

The academic editor has obtained 3 reviews, which are included below. The editor requests that you constructively address the concerns of the reviewers in a revised manuscript. In particular, the following issues should be addressed.

1. The concept of a regenerative window closing/opening versus a delay in closing should be explained. If a window manipulated with DCC/cAMP/Rac1 "reopened" it means that it was initially closed. However, there could be a delay or a decline in regenerative potential. Data supporting regenerative potential after 3 dpf could be useful to clarify this point.

2. Provide more evidence regarding the damage caused by photoablation by doing and showing "mock" control ablations outside the axons.

3. The localization of glia should be characterized and the direct involvement of glia should be further characterized by doing glia ablation and comparing to "mock" ablation outside the glia.

4. Study whether DCC and Rac1 manipulation alter regenerative potential.

5. Investigate whether overexpression of secreted netrin from glia or other tissues can prematurely close the regenerative window.

**Journal Requirements:**

https://journals.plos.org/plosgenetics/s/submission-guidelines#loc-parts-of-a-submission

3) We noticed that you used the phrase 'data not shown' in the manuscript. We do not allow these references, as the PLOS data access policy requires that all data be either published with the manuscript or made available in a publicly accessible database. Please amend the supplementary material to include the referenced data or remove the references.

4) We do not publish any copyright or trademark symbols that usually accompany proprietary names, eg ©,  ®, or TM  (e.g. next to drug or reagent names). Therefore please remove all instances of trademark/copyright symbols throughout the text, including:

- © on page: 24

- ® on page: 24.

5) We notice that your supplementary Figures are included in the manuscript file. Please remove them and upload them with the file type 'Supporting Information'. Please ensure that each Supporting Information file has a legend listed in the manuscript after the references list.

Potential Copyright Issues:

i) Figures 7B, and 8B. Please confirm whether you drew the images / clip-art within the figure panels by hand. If you did not draw the images, please provide (a) a link to the source of the images or icons and their license / terms of use; or (b) written permission from the copyright holder to publish the images or icons under our CC BY 4.0 license. Alternatively, you may replace the images with open source alternatives. See these open source resources you may use to replace images / clip-art:

7) We note that your Data Availability Statement is currently as follows: "All data related to this manuscript is available in the enclosed material.". Please confirm at this time whether or not your submission contains all raw data required to replicate the results of your study. Authors must share the “minimal data set” for their submission. PLOS defines the minimal data set to consist of the data required to replicate all study findings reported in the article, as well as related metadata and methods (https://journals.plos.org/plosone/s/data-availability#loc-minimal-data-set-definition).

8) Please amend your detailed Financial Disclosure statement. This is published with the article. It must therefore be completed in full sentences and contain the exact wording you wish to be published.

2) If any authors received a salary from any of your funders, please state which authors and which funders..

**Reviewers' comments:**

Reviewer's Responses to Questions

**Comments to the Authors:**

Reviewer #1: The authors of this manuscript investigate DRG central axon regeneration after severing, and specifically the mechanisms that close the temporal window during which regeneration can take place. They find that zebrafish larvae can regenerate severed DRG central axons on day 2 of development, but lose that ability on day 3. Using a combination of genetic and pharmacological approaches, they provide evidence that the regeneration window closes because netrin signaling, most likely from adjacent glial cells, prevents appropriate formation of the invadopodia required by severed DRG central axons to enter the spinal cord at the dorsal root entry zone. They examined steps in the netrin signaling pathway and showed that netrin levels increase in glia in the path of regenerating axons at the time when regenerative ability is diminishing, and that expression of the DCC netrin receptor in DRG neurons also increases in the same time frame. Surprisingly, they found that overexpressing DCC in these neurons at a time when the regenerative window is still open does not prematurely close the regeneration window. They did not test whether overexpressing netrin in glia would prematurely close the regenerative window because glia do not reorganize until 3 days. However, since netrin is a secreted molecule, this seems like a missed opportunity to address one prediction of their netrin signaling hypothesis by misexpressing netrin in other cell types, or implanting netrin expressing beads. Instead, they targeted netrin signaling downstream of DCC in dcc heterozygotes, using agonists and antagonists of cAMP and Rac1 and found that upregulation of this pathway decreases and down regulation of this pathway promotes invadopodia persistence at the dorsal root entry zone, and that this occurs through DCC acting within DRG neurons. They also found that antagonizing DCC signaling enabled DRG axons to regenerate following closure of the regeneration window, as assayed by calcium responses to cold water sensory stimuli and resulting behavioral responses that require intact DRG to CNS circuitry.

The study follows nicely from previous work in the Smith lab. It is generally well done and well presented, although there are a few issues that remain to be addressed.

1) It would be very helpful to clarify which of the processes described in the Introduction (lines 72-84) are referring to mammals, as at the beginning of the paragraph, and which are referring to zebrafish, as at the end of the paragraph. More specifically, are the invadopodia studies cited only from zebrafish? In the following paragraph it would also be useful to state explicitly that this work is from zebrafish.

2) This is a complicated study with many moving parts. It would be really helpful to have a graphical abstract or a summary diagram that puts all the pieces to together so that readers can see the entire context, including diagramatic representations of the treatments used to assess the role of DCC downstream components.

3) How many DRG neurons are present per DRG at the time of central axon severing in this study? In movie 6 it looks like there are at least two axons and that they go in different directions and perhaps do different things. Do all of the DRGs used in this study have the same number of neurons? Do all of the neurons within a DRG develop within the same time frame? Or are there pioneering neurons and ones that develop later? Do all of the axons from neurons within an individual DRG follow a similar regenerative path? My understanding is that zebrafish DRGs add neurons progressively. I realize that the authors have only investigated the first few days of development, but they might think about, and perhaps discuss, the regenerative capacity of DRG neurons that are added later. In other words, does the regenerative window close early so that only the first-developing DRG neurons can send central axons into the CNS following injury, or is there some kind of sliding window so that later-developing DRG neurons are also able to regenerate during later windows?

4) It would be very helpful to mark where the cut occurs on at least the first figure showing the axon severing.

5) Figures and figure panels are not all in order (eg some later panels are described before earlier panels of the same figure). And some panels don’t seem to be mentioned in the text.

6) Many of the figure legends should provide more description. For example, in Figure 3F, what do DRG 3, DRG 4, etc. signify?

7) It’s difficult to compare Figures 5B and 5C since they were made with different graphing formats.

8) In Figure 5F, why are the cortactin only IHC panels (two on the right) offset from the others? Is that the same spot as in the second panel from the left? If so, why are they not aligned in all of the panels? If not, what are we seeing?

9) What is the rationale for cutting DRG axons from multiple segments in the behavioral experiments?

10) In the discussion the authors try to draw some distinctions between development and regeneration. However, they don’t carefully define the developmental time frame they focus on, and therefore some of their language seems confusing.

11) There are a number of instances in the manuscript where the subject and verb do not agree. And there are some typos that warrant spell checking.

Reviewer #2: The manuscript by Hammer and Smith examines regeneration of the central axons of DRG/somatosensory neurons following laser axotomy in the larval zebrafish spinal cord. The work uses a combination of genetics, chemical genetics and live imaging to investigate the role of glia and a DCC signaling axis -- which previously described by the authors as functioning during the development of this system (Kikel-Coury et al., 2021) -- in DRG axon regeneration.

The work is novel and will be of interest to researchers in the neuroscience and regeneration fields. While the genetic and epistatic experiments are strengths of the manuscript, as discussed below, there are several other areas that should be strengthened/revised prior to publication.

Major comments

* Although the authors claim that the photoablation system used does not cause excessive damage to surrounding tissue, it was difficult to assess this claim. For example in Fig. 1D there is no control image to compare against. And in Movie S1 there appear to be: 1) examples of cell death; 2) CNS axon regeneration across the lesion site; and 3) increased sox10+ cells in the lesioned area following ablation (all of which could result from non-specific laser damage). As there is not a control video presented here it is unclear whether these are also occurring in the absence of photoablation.

* The involvement of glia in the regenerative response remains speculative. The authors propose that glia reorganize following axotomy at 3 dpf but not 2 dpf and that they inhibit regeneration at 3 dpf. However, there are several issues with the analysis. First, as shown in Fig. 2A it appears that the main glial reorganization is between 2 and 3 dpf not after the 3 dpf axotomy as claimed. To address this, in Figure 2C, the authors should compare the two glial values to each other. They should also calculate the relative change in glial position over the two experimental windows (t140-t0) to address the relative amount of reorganization. Second, the glia ablation experiment is lacking an essential control, which is an ablation with the same laser exposure as shown in 2D (targeted outside the glia). Without this, it is unclear whether the increased axon regeneration is due to loss of the glia or increased laser damage (which could cause increased ROS. See e.g., Rieger S, Sagasti A (2011) DOI: 10.1371/journal.pbio.1000621). Third, a limitation of the glial reorganization analysis is that they are only observing nuclear position and not membrane morphology. This could be mentioned in the “limitations” subsection of the discussion.

* The cell-specific manipulations of DCC and Rac1 are based on the expression of UAS-driven constructs in the sox10:Gal4 line. Because sox10 is expressed in several cell types (somatosensory neurons, surrounding glia, neural crest precursors) in the area (Zhu et al. (2019) DOI: 10.1016/j.cell.2019.08.001), in the absence of a more specific driver, the authors should temper/clarify their language around the cell specificity claims.

* Statistics. The numerous P-values displayed on several of the plots were distracting (e.g., Figs 7D,E, 8D) – the authors may consider moving some of this reporting information into the supplement to focus on the data display. Were the statistical tests corrected for multiple comparisons? The use of the “Fisher’s Exact t-test” was curious for two reasons: 1) it is not a t-test; and 2) it is only for 2x2 tables but appears to have been used in cases with three categories (e.g., Figs 1E, 4B).

Minor comments

* A developmentally regulated diminished capacity for skin reinnervation was previously described for larval somatosensory neurons by O’Brien et al. (2009) DOI: 10.1016/j.cub.2009.10.051. It would be important to contextualize the current results relative to this previous study.

* It would be helpful to contextualize the results of the first results section relative to known events that are happening at 2 vs 3 dpf for zebrafish DRG neurons. For example, when does growth of the central axon begin/terminate?

* Figure legends do not indicate what the error bars represent (standard deviation, standard error of the mean, etc).

* Figure 1 and lines 110-116. The panels are referred to out of order in the text. Figure 1B is not mentioned in the text.

* Figure 1A. Consider adding labels or text to indicate the axes (e.g., Dorsal/Ventral)

* Figure 1C and Figure 4D,F. Missing scale bars and hard to see – consider adding a higher magnification inset.

* Figure 3C,D. Descriptions are reversed in figure legend.

* Figure 3D and lines 154-156. Netrin looks like it is broadly increased rather than specifically dorsal. Considering quantifying and comparing the lateral and ventral domains.

* Line 158. It is unclear how the “midpoint of regeneration” was defined.

* Figure 3B. A small schematic may be helpful for explaining the experimental design.

* Lines 173-174. The involvement of netrin1b remains speculative and the text on L173-4 “both dcc and netrin1b are organized in the tissue to close the regenerative period at 3 dpf.” was unclear/imprecise.

In the text, the authors should state: What is the nature of the ZM130198 allele? Why was this allele chosen relative to other available dcc alleles?

* Figure 4G,H. What was the treatment/background for these panels?

* Lines 226-228. The rationale for suspecting a failure in invadopodia formation was unclear. Seems like the movies/images don’t support a failure at the DREZ but earlier. Consider revising the text to better set up logic/hypothesis

* Figure 5A and Figure 6. Typo on y-axis labels “Intesnity”.

* Figure 5B,C. What are the n values?

* Figure 7C. What does the “lz-score” label signify?

* Supplemental movies lack scale bars.

* Reference 61 is a duplicate of 45 and reference 70 is incomplete.

Reviewer #3: This manuscript by Hammer and Smith interrogates the closure of a window of regenerative plasticity in DRG neurons of Zebrafish. Understanding the molecular mechanisms that govern the switch from a plastic state during development that is conducive to regenerative and growth, to a ‘post-developmental’ state that is not permissive is a key question in neurobiology. Understanding how this state change is regulated will be important to develop therapies for neurodevelopmental disorders as well as disease and injuries to the nervous system.

The authors characterise an in vivo model of regeneration using the DRG neurons of zebrafish and establish that they undergo robust regeneration that rapidly declines over a 24hr window between 2dpf and 3dpf. They identify that glial cell populations also reorganise during this time window and identify netrin signalling as a key regulator of this process, with glial cells the likely source. Using a combination of transgenic lines and pharmacological manipulations they present evidence that netrin functions via DCC>cAMP>Rac1 in DRG neurons to regulate regenerative potential and the closing of this regenerative period.

The authors using live imaging to characterise the development of distinct invadopodia and actin dynamics that correlate with regenerative potential that are regulated by this signalling axis. The show that regenerative potential correlates with functional recovery and that manipulating the netrin/DCC/rac pathway can restore both spinal motorneuron activity and behaviour following injury.

This is an important and interesting study that will be of interest to the field of axonal regeneration, glial cell biology and axonal guidance.

There are several elements that I think can be refined to improve the manuscript and solidify the main findings.

Main points:

1. The decline in regenerative potential is characterised as a window, that can be manipulated with DCC/cAMP/rac1 signalling to be ‘re-opened’. I don’t think the data fully supports this notion. First, in Figure 1 the regenerative decline is characterised across 2dpf, 3dpf and 5dpf. All other data in which the window is re-opened is only done at 3dpf. This supports the notion that the decline is prevented until after 3dpf. But does not provide any information about any regenerative potential after 3dpf. This is important to distinguish if these animals have no decline in potential, or a delay. Moreover, to support the notion that a window is ‘reopened’ – something needs to close. Ie. a condition in which 3dpf there is limited regeneration and then a manipulation to restore regeneration in 5dpf. This is a major claim of the manuscript. I think the language needs to be revisited if the data is not supporting this concept.

2. The quantification of glia between 2 and 3 dpf is very interesting and in the images and videos presented there is a stark difference. I would be very interested to see the localisation of glial cells at these ages in uninjured animals to further characterise this relationship. At 3dpf it appears that some glia flank the base of the central axon. Is this a response to injury, or the normal cell position at this age? Furthermore, what is the glia phenotype in the DCC/cAMP/Rac1 manipulations? Do they still localise around the axon in animal where the regenerative potential is regained?

3. The experiments using cell-specific manipulations of DCC and Rac1 are only characterised with invadopodia imaging and functional assays. Demonstrating that these manipulations alter the regeneration phenotype would solidify this data.

Minor points:

1. A scheme of the orthogonal view shown in Figure 1C (similar to the inset in Figure 1A) might help with understanding the anatomy of the phenotype and how regeneration was scored. It took me a little while to wrap my head around this the first time I read it.

2. Many of the figures do not recapitulate the detail captured in the live movies. In particular, the lifeact videos are stunning – with lots of information and dynamics that are not represented in the static figure. This is challenging for all live imaging. But I would recommend revising these figures to try and capture more of the dynamics. The actin traces are useful – but the image panels are very small and it’s difficult to see the actin structures in the growth cone. The same is true for the images of regeneration in Figure 1.

3. Related to the above point. It’s not entirely clear what primary phenotype is in 3dpf injured control animals vs DCC manipulations. In Figure 1 and Movie S2 it appears that at 3dpf the central axon never really makes it out of the glial surrounding region just dorsal to the injury. But in the lifeact images it appears that the axon extends – but is not stabilised and retracts. These are important details. Especially given the correlative data in relation to glia as a source of netrin to inhibit regeneration.

**Have all data underlying the figures and results presented in the manuscript been provided?**

Reviewer #1: Yes

Reviewer #2: None

Reviewer #3: Yes

PLOS authors have the option to publish the peer review history of their article (what does this mean? ). If published, this will include your full peer review and any attached files.

**Do you want your identity to be public for this peer review?** For information about this choice, including consent withdrawal, please see our Privacy Policy .

Reviewer #1: No

Reviewer #2: No

Reviewer #3: No

**Figure resubmission:**
---

## [Decision Letter · Decision Letter 1]

13 Jan 2026

Dear Dr Smith,

We are pleased to inform you that your manuscript entitled "The regenerative period of somatosensory nerves is closed by a DCC signaling axis" has been editorially accepted for publication in PLOS Genetics. Congratulations!

Before your submission can be formally accepted and sent to production you will need to address and correct the minor suggestions from reviewers 1 and 3 (see below) and complete our formatting changes, which you will receive in a follow up email. Please be aware that it may take several days for you to receive this email; during this time no action is required by you. Please note: the accept date on your published article will reflect the date of this provisional acceptance, but your manuscript will not be scheduled for publication until the required changes have been made.

Yours sincerely,

Benjamin Podbilewicz

Academic Editor

PLOS Genetics

Fengwei Yu

Section Editor

PLOS Genetics

Aimée Dudley

Editor-in-Chief

PLOS Genetics

Anne Goriely

Editor-in-Chief

PLOS Genetics

BlueSky: @plos.bsky.social

Comments from the reviewers (if applicable):

Reviewer's Responses to Questions

**Comments to the Authors:**

Reviewer #1: The authors have done a nice job responding to most of my earlier comments, and I am satisfied with their revisions. Here I point out a few minor things that remain to be fixed:

Subject verb agreement:

1) lines 21-22 should be either “zebrafish larvae as a model” or “the zebrafish larva as a model”

2) lines 44-45 should be “the regenerative period for these nerves is closed” or “the regenerative periods for these nerves are closed”

3) line 72 should be “the concept of critical periods in neural development parallels”

4) line 92 should be “regeneration of mammalian centrally-projection axons is”

5) line 160 should be “glial nuclear position”

6) line 182-183: should be “re-organization of glia surrounding the DRG in 3 dpf animals takes part”

Other:

1) What does “it” in line 81 refer to

2) line 149 satellite glial cells

3) line 166 glia were an average of

4) lines 279-280 do you mean not DCC mutation, nor treatment…….

5) line 983 where are the yellow arrows?

6) line 1025 much of the figure legend below this point is in italics

Reviewer #2: The authors clearly addressed our questions and concerns through new experimental controls/analysis and changes to the text.

Reviewer #3: The authors have addressed all points and concerns from the original submission. I think these experiments have significantly improved the manuscript. Especially around the notion of a ‘regenerative window’ and the concept that this can be re-opened.

Minor comments:

1. In response to reviewer 1, point 2, the authors state that they have included a graphical abstract/summary diagram. I cannot see this in the documents provided and there is no reference to this in the point-by-point response.

2. There is a reference to Figure 2E on line 186. Should this be Figure 2F?

3. There is a reference to the ‘regenerative period’ in line 233. I would suggest revising this to ‘regenerative window’. This should be consistent throughout the manuscript.

**Have all data underlying the figures and results presented in the manuscript been provided?**

Reviewer #1: Yes

Reviewer #2: None

Reviewer #3: Yes

PLOS authors have the option to publish the peer review history of their article (what does this mean? ). If published, this will include your full peer review and any attached files.

**Do you want your identity to be public for this peer review?** For information about this choice, including consent withdrawal, please see our Privacy Policy .

Reviewer #1: No

Reviewer #2: No

Reviewer #3: No

**Data Deposition**

http://datadryad.org/submit?journalID=pgenetics&manu=PGENETICS-D-25-00917R1

**Press Queries**

---

## [Editor Report · Acceptance letter]

PGENETICS-D-25-00917R1

The regenerative period of somatosensory nerves is closed by a DCC signaling axis

Dear Dr Smith,

We are pleased to inform you that your manuscript entitled "The regenerative period of somatosensory nerves is closed by a DCC signaling axis" has been formally accepted for publication in PLOS Genetics! Your manuscript is now with our production department and you will be notified of the publication date in due course.

With kind regards,

Anita Estes

PLOS Genetics

On behalf of:
